

# Long-live High Frequency Gravity Waves in Atmospheric Boundary Layer: Observations and Simulations

Mingjiao Jia[1,*], Jinlong Yuan[1,2,*], Chong Wang[1,2], Haiyun Xia[1,2], Yunbin Wu[2], Lijie Zhao[2], Tianwen Wei[2], Jianfei Wu[2], Lu Wang[2], Sheng-Yang Gu[3], Liqun Liu[4], Dachun Lu[5], Rulong Chen[5], Xianghui Xue[2], Xiankang Dou[3]

[1]Glory China Institute of Lidar Technology, Shanghai, 201315, China
[2]CAS Center for Excellence in Comparative Planetology, University of Science and Technology of China, Hefei, 230026, China
[3]School of Electronic Information, Wuhan University, Wuhan, 430072, China
[4]Anqing Meteorological Bureau, China Meteorological Administration, Anqing, 246001, China
[5]Technical Support Center for Atmosphere Observation, Anhui Meteorological Administration, Hefei, 230031, China
[*]These authors contributed equally to this work.

*Correspondence to*: Haiyun Xia (hsia@ustc.edu.cn)

**Abstract.** A long-live gravity wave (GW) in atmospheric boundary layer (ABL) during a field experiment in Anqing, China (116 °58′ E, 30 °37′ N) is analysed. Persistent GWs over 10 hours with periods ranging from 10 to 30 min in the ABL within 2 km height are detected by a coherent Doppler lidar from 4 to 5 in September 2018. The amplitudes of the vertical wind due to these GWs are about 0.15~0.2 m s$^{-1}$. The lifetime of the GWs is more than 20 wave cycles. There is no apparent phase progression with altitude. The vertical and zonal perturbations of the GWs are apparent quadrature with vertical perturbations generally leading ahead of zonal ones. Based on experiments and simplified 2-Dimensional Computational Fluid Dynamics (CFD) numerical simulations, a reasonable generation mechanism of this persistent wave is proposed. A westerly low-level jet of ~5 m s$^{-1}$ exists at the altitude of 1~2 km in the ABL. The wind shear around the low-level jet lead to the wave generation in the condition of light horizontal wind. Furthermore, a combination of thermal and Doppler ducts occurs in the ABL. Thus, the ducted wave motions are trapped in the ABL with long lifetime.

## 1 Introduction

The atmospheric boundary layer (ABL) is the most important atmospheric environment affecting the human life. Gravity waves (GWs) and corresponding physical processes have important impacts on synoptic systems, atmospheric models, aircraft take-off and landings in the ABL (Clark et al., 2000; Fritts and Alexander, 2003; Holton and Alexander, 2000; Sun et al., 2015b). GWs are ubiquitous in the atmosphere and usually generated from topography, convection, wind shear, jet streams, frontal systems and other sources in the troposphere (Banakh and Smalikho, 2016; Blumen et al., 1990; Chouza et al., 2016; Fritts and Alexander, 2003; Plougonven and Zhang, 2014; Pramitha et al., 2015; Toms et al., 2017; Wu et al., 2018). As propagating upward, GWs will become unstable or break when a critical layer is encountered. On the one hand,



these breaking GWs would cause clear air turbulence, which is hazardous to aircraft as it is neither visible to pilots nor detectable by standard on-board radars (Bramberger et al., 2018; Clark et al., 2000; Ralph et al., 1997). On the other hand, these processes will lead to transportation of energy and momentum from lower atmosphere to upper atmosphere, and thus affect the coupling between lower atmosphere and upper atmosphere, as well as dynamic and thermal structure of the global

atmosphere (Fritts and Alexander, 2003; Holton and Alexander, 2000). Therefore, GW plays a key role in aviation safety, weather forecast and climate models.

High frequency GWs with periods less than one hour contribute most to transporting momentum into upper atmosphere (Fritts and Vincent, 1987). However, these GWs and their sources are difficult to be resolved in global general circulation models due to smaller spatial and temporal scales. Only mesoscale and larger scale GWs can be resolved in global

atmospheric models (Preusse et al., 2014; Wu et al., 2018). GW parameterizations are always used in global models to increase their reliability and precision (Fritts and Alexander, 2003). Thus there is requirement to improve our understanding of high frequency GWs and their sources.

However, wave motions in the ABL are usually difficult to be detected due to the contaminations from strong turbulence. Therefore, most wave motions are observed in the stably stratified ABL (Banakh and Smalikho, 2016; Fritts et al., 2003;

Mahrt, 2014; Sun et al., 2015a; Sun et al., 2015b; Toms et al., 2017). These wave motions can be maintained more than a few periods if atmospheric wave ducting properties are present, while such monochromatic waves are infrequently observed (Mahrt, 2014; Toms et al., 2017). In addition, due to the capabilities of ground-based measurements, most of these previous studies are limited to the surface layer within tens or hundreds of meters near the ground, not the whole ABL.

Numerous instruments have been utilized to detect wave motions in the ABL. Fixed point measurements on a tower or

surface (Einaudi and Finnigan, 1981; Finnigan and Einaudi, 1981; Poulos et al., 2002; Sun et al., 2015a; Sun et al., 2004), in-situ measurements on mobile platform such as balloon (Corby, 1957), aircraft (Fritts et al., 2003; Kuettner et al., 2007), remote sensing measurements such as sodar (Beran et al., 1973; Hooke and Jones, 1986; Lyulyukin et al., 2015), radar (Cohn et al., 1997; Cohn et al., 2001) and lidar (Chouza et al., 2016; Mayor, 2017; Neiman et al., 1988; Newsom and Banta, 2003; Poulos et al., 2002; Witschas et al., 2017) are widely used in recent decades. All these techniques are sensitive to only a

certain portion of the wave spectra and wave characteristics, given a limited spatial and temporal range. Among these instruments, lidar can make measurements alone with sufficient long detection range, multi scanning mode, high temporal/spatial resolution. Recently, a micro-pulse coherent Doppler lidar (CDL) is developed to measure wind field with temporal resolution of 2 s and spatial resolution of 60 m in ABL (Wang et al., 2017). Wave motions such as high frequency GWs can be revealed from the vertical wind measured by this lidar in the whole ABL.

Numerical simulations are also used to study GWs. Mesoscale and large scale GWs can be resolved in high spatial and temporal resolution models such as Whole Atmosphere Community Climate Model (WACCM) and Weather Research and Forecasting (WRF) (Wu et al., 2018). For high frequency GWs with smaller scales, high resolution Computational Fluid Dynamics (CFD) simulations have been used in recent years (Chouza et al., 2016; Watt et al., 2015). CFD simulation is able to resolve the flow field at different spatial scales, ranging from mesoscale of ~200 km to indoor environment of ~10 m



(Berg et al., 2017; Fernando et al., 2018; Mann et al., 2017; Remmler et al., 2015; Ren et al., 2018; Toparlar et al., 2015; Toparlar et al., 2017; Vasiljević et al., 2017; Watt et al., 2015). With the help of CFD simulation, the generation mechanism and characteristics of GWs can be resolved, as well as the subsequent evolutions of GWs.

In this paper, we report long-live high frequency GWs in the whole ABL detected by the CDL. The characteristics and the generation mechanism are analysed using experiments and CFD simulations. Section 2 describes the field experiments and instruments used in this study. Section 3 presents the observational results and corresponding analysis of the GWs. The CFD model and simulation results are described and discussed in Sect. 4. Section 5 gives a discussion of the generation mechanism of the persistent GWs. Finally, the conclusion is drawn in Sect. 6. If not specified, local time is used in this paper.

## 2 Experiments and instruments

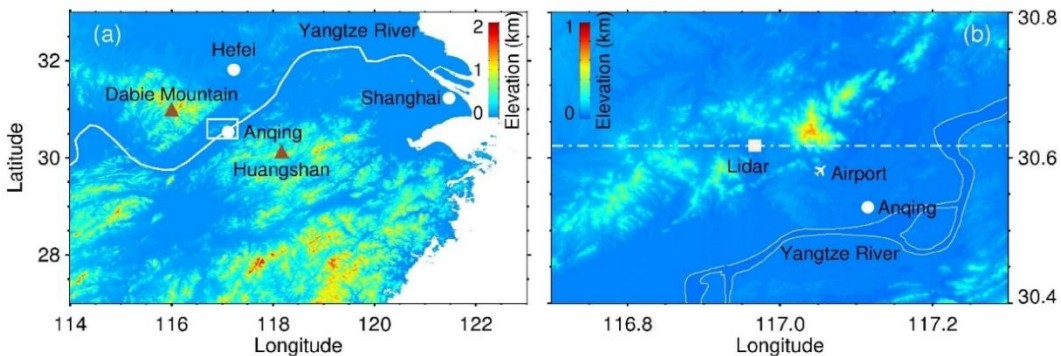

**Figure 1.** (a) Terrain elevation map. (b) Zoom over Anqing station in the white hollow rectangle in (a). The computational domain is roughly along the white dash dotted line in latter numerical simulations.

A field experiment is conducted to study the generation mechanism of GWs by the CDL in National Meteorological Observing Station of Anqing (116°58′ E, 30°37′ N) from 16 August to 5 September 2018. Anqing is located near Yangtze River and between Huangshan (118°10′ E, 30°08′ N) in the southeast and Dabie Mountain (115°~117° E, 30°~32° N) in the southwest as shown in Fig. 1a. The station is surrounded by hills with relative elevation of 200~600 m as shown in Fig. 1b. An airport is located in the southeast of the station.

### 2.1 Coherent Doppler wind lidar

A compact micro-pulse CDL working at eye-safe wavelength of 1.5 μm is used in this study. The pulse duration and pulse energy of the laser are 300 ns and 110 μJ, respectively. A double D-shaped telescope is employed. The absolute overlap distance and blind distance are ~1 km and 60 m, respectively. This lidar has full hemispheric scanning capability with the rotatable transmitting and receiving system. Benefiting from the coherent detection, this lidar can perform all-day measurement of radial wind speed based on the Doppler effect. Compared with traditional lidars, this CDL is small in size





and robust in stability due to the all-fiber configuration. More details of this lidar are described in Wang et al. (2017). The key parameters of the CDL are listed in Table 1.

**Table 1.** Key Parameters of the CDL

| Parameter | Value |
| --- | --- |
| Wavelength | 1548 nm |
| Pulse Duration | 300 ns |
| Pulse Energy | 110 μJ |
| Repetition frequency | 10 kHz |
| Diameter of telescope | 80 mm |
| Spatial resolution | 60 m |
| Temporal resolution | 2 s |
| Maximum range | 15 km |
| Azimuth scanning range | 0 - 360 ° |
| Zenith scanning range | 0 - 90 ° |

Wind filed is composited by pointing the rotatable scanner at three directions during the experiment. Firstly, the laser beam

is pointed at two orthogonal azimuths sequentially, north and west with a zenith angle of 30 °. Then, the laser beam is pointed vertically upward. In each direction, the measurement duration is set to 10 s during this experiment. The full period of the measurement cycle is 41 s. The observational results, such as vertical and horizontal wind components, and carrier to nose ratio (CNR) in the vertical beam, are shown in Appendix A. The blank areas without measurements are owing to the rainy summer. For example, the No. 18 Typhoon Rumbia passed by around 17 August 2018. To guarantee the precision of the

wind measurements, the data with CNR less than -35 dB is abandoned (Wang et al., 2017; Wang et al., 2019).

## 2.2 Radiosonde

National Meteorological Observing Station of Anqing is one of the 120 operational radiosonde stations in China (excluding Hong Kong and Taiwan) (Li, 2006). China Meteorological Administration has deployed an L-band (1675 MHz) sounding system in this station. Air temperature, pressure, relative humidity and wind from the ground to middle stratosphere can be

measured twice a day at 07:15 and 19:15 by this sounding system, which combines a GTS1 digital radiosonde with a secondary wind-finding radar. Previous studies confirmed the accuracy measured by this type of radiosonde (Bian et al., 2010).

## 2.3 ERA5 reanalysis data

ERA5 is the fifth generation of ECMWF (European Centre for Medium-Range Weather Forecasts) atmospheric reanalysis of

the global climate. ERA5 reanalysis assimilates a variety of observations and models in 4-dimensional. The data has 137 levels from the surface up to 80 km altitude and horizontal resolution of 0.3 ° for both longitude and latitude (Hersbach and





Dee, 2016). The hourly temperature data from high resolution realisation sub-daily deterministic forecasts of ERA5 is used to calculate buoyancy frequency near the station in latter analysis in this study.

## 3 Observations and analysis

### 3.1 The long-live GWs

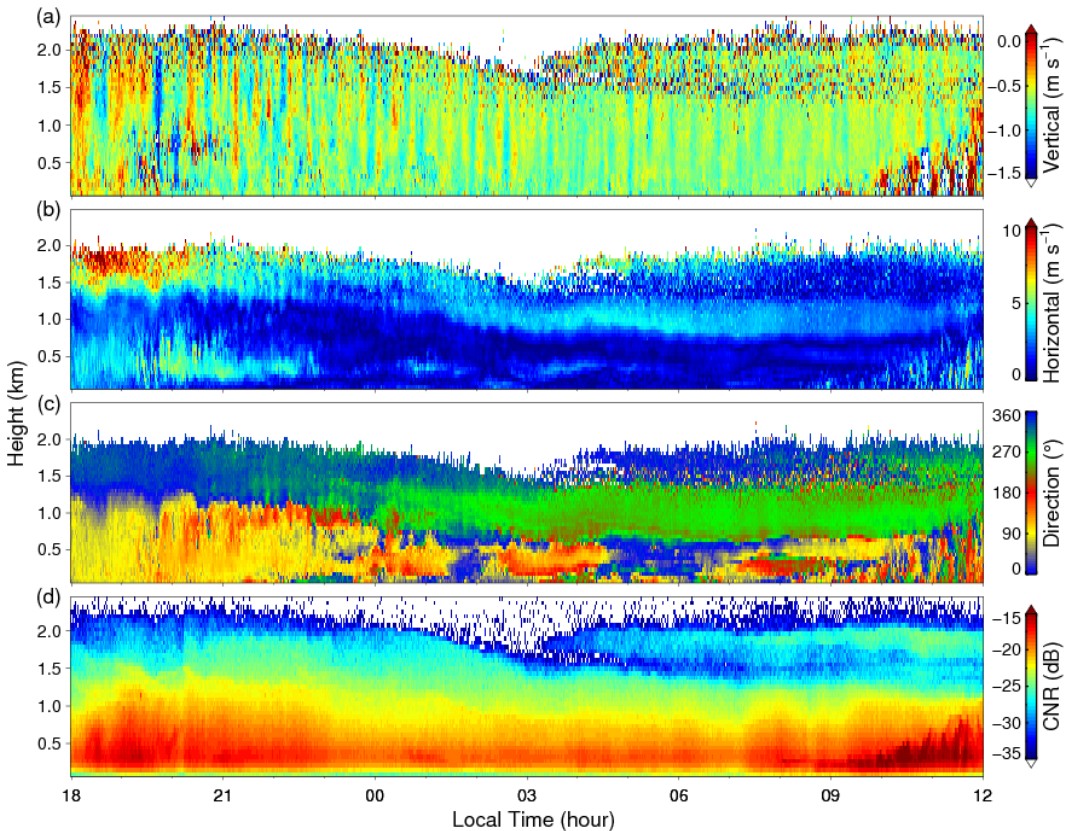

**Figure 2.** Height-time cross sections of the (a) vertical wind, (b) horizontal wind speed, (c) horizontal wind direction and (d) CNR in vertical obtained by the CDL from 4 to 5 September 2018. The direction is defined as the angle clockwise from the north.

Figure 2a shows the persistent wave motions in the vertical wind longer than 10 hours in the ABL between 4 and 5 September 2018. These waves exist more than 20 periods and then dissipated during the evolution of convective ABL in the morning on 5 September 2018. The corresponding horizontal wind speed and wind direction are shown in Fig. 2b and 2c. Two weak low-level jets are observed at heights of about 0.5 and 1.5 km. The lower easterly jet stream lasts only a few hours with speed of about 5 m s$^{-1}$, while the higher jet stream exists during the whole lifetime of the wave motions. The speed of the higher jet stream is about 10 m s$^{-1}$ and then decreases to about 3~5 m s$^{-1}$ after 21:00. The corresponding direction of this





northerly jet stream is also changed to westerly. The CNR from the vertical beam is shown in Fig. 2d, which varies slowly with time and nearly stratified in altitude. Thus the ABL is stably stratified as the CNR represents the aerosol concentration. The periods of these wave motions are typically about 10~30 minutes. The temporal profiles of average vertical wind between 600 m and 1000 m is plotted in Fig. 3a. Oscillations of vertical wind can be seen clearly. The amplitudes of these

wave motions are about 0.2 m s$^{-1}$ before 03:00 and then decreases to about 0.15 m s$^{-1}$ while the periods extended after 04:00. The wavelet power spectrum of the vertical wind in Fig. 3a is shown in Fig. 3b by using the Morlet mother wavelet. There are obvious waves with periods of 15~25 min before 03:00 and waves with periods of 20~30 min after ~04:30. Relatively weak waves with periods of about 10 min are also observed between 03:00 and 05:00. These wave motions could be regarded as quasi monochromatic waves as the periods varies within the range of 15~30 min. The change of periods may be

in relation to the changes of the background ABL, such as the height of upper jet stream.

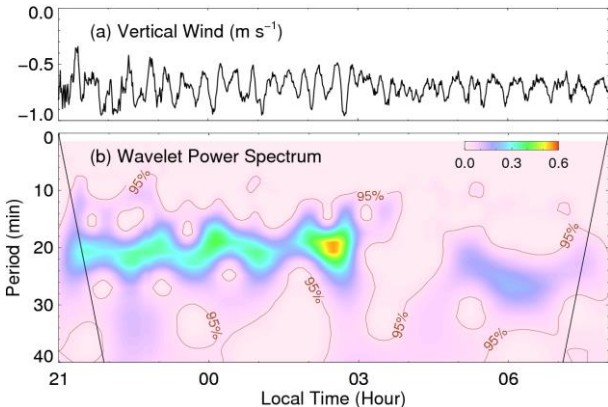

**Figure 3.** (a) Mean vertical wind between 600 m and 1000 m. (b) Corresponding wavelet power spectrum of the vertical wind in (a). The brown contours indicate significance level of 95%. The black solid lines represent the Corn-of-Influence.

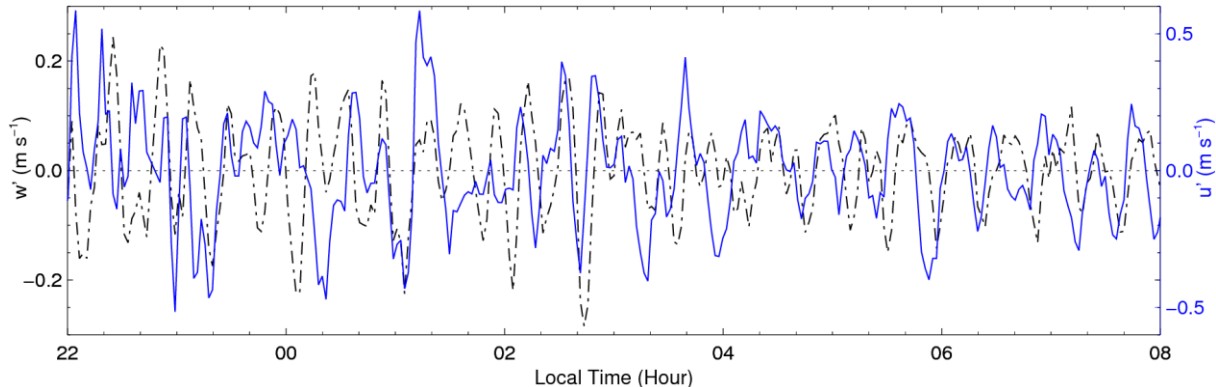

**Figure 4.** Perturbations of vertical wind **w′** (black dash dotted, left axis) and zonal wind **u′** (blue solid, right axis) obtained between 600 m and 1000 m altitude from 22:00 on 4 September to 08:00 on 5 September 2018.

Zonal wind can be derived from the horizontal wind speed and direction. The height averaged perturbations of vertical wind **w′** and zonal wind u′ between 600 m and 1000 m altitude are shown in Fig. 4. First, the raw vertical/zonal wind are



averaged with altitude between 600 m and 1000 m. Second, the temporal profile of averaged vertical/zonal wind is smoothed by a 1-hour window as the background. Thirdly, the background is subtracted to remove the trend. Finally, the perturbation is smoothed by averaging the adjacent three points to reduce high frequency noises. It is obvious that the wave motions also exist in the horizontal wind. The periods of the zonal perturbations are similar to that of vertical perturbations. Specifically, vertical and zonal perturbations are apparent quadrature with vertical perturbations $\mathbf{w}'$ generally leading zonal perturbations $u'$, especially after 02:00. Note that the wave motions exhibit highly coherent vertical motions with no apparent phase progression with altitude as shown in Fig. 2a. These characteristics of these wave motions indicate ducting wave structures within the ABL (Fritts et al., 2003).

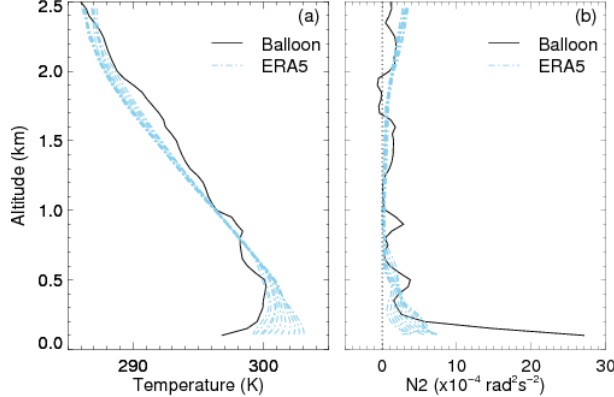

**Figure 5.** (a) Temperature profiles from radiosonde (black solid) and ERA5 (skyblue dot dashed) during the wave motions. (b) Corresponding buoyancy frequencies.

Temperature profile measured by the radiosonde lifted on a balloon at about 07:15 on 5 September 2018 and hourly temperature profiles from ERA5 between 22:00 on 4 September and 08:00 on 5 September are shown in Fig. 5a. An inversion layer is observed under altitude of ~500 m. The corresponding squares of buoyancy frequency ($N^2$) are plotted in Fig. 5b. Maxima values of $N^2$ larger than $5 \times 10^{-4} \, rad^2 s^{-2}$ appear in the inversion layer from both radiosonde and ERA5, indicating a strongly stratified stable boundary layer near ground. Between ~600 and ~2000 m altitude, the values of $N^2$ are so small that close to zero, even negative at 1800~2000 m from radiosonde. There are no obvious wave motions above this region between 04:00~09:00 in the vertical wind as shown in Fig. 2a, though with large enough CNR in Fig 2d. These results also hint a thermal ducting between the ground and about ~2000 m, in which the wave motions are trapped, especially under the inversion. This is why such wave motions have a long lifetime longer than 20 periods. The buoyancy periods from Fig. 5b are typically 2~10 min. Since the background wind speeds are relatively small, less than ~10 m s⁻¹, we neglect Doppler effects here. These wave motions should be GWs instead of internal acoustic waves. Therefore, these waves are suggested to be ducted gravity waves trapped in the ABL.





## 3.2 Background wind

There are complex relationships between GWs and background wind conditions. Submeso wavelike motions, that any nonturbulent motions on horizontal scales smaller than 2 km and with periods of tens of minutes, are primarily generated in very weak winds in nocturnal boundary layer (Mahrt, 2014). Noted that the wind speed and wind shear from 4 to 5 in September 2018 are weakest during the whole field experiment in Fig. A2. In order to understand the relationship between GWs and background wind, dominant GW are identified using the method described in Appendix B in each temporal-spatial window. Here we focused on high frequency GWs with periods of 10~50 min during the whole experiment. Therefore, a window of 4-hour length and 200-m height, and shifted in steps of 1 hour temporally and 100 m vertically is used. For the background wind, mean horizontal wind speed and wind direction in the central 1-hour of each window are easily obtained. Wind shear $S$ can be calculated from the vertical profiles of mean horizontal wind speed, in which $S$ is defined as below:

$$S = ((dU/dz)^2 + (dV/dz)^2)^{1/2} \tag{1}$$

where $U$ ($V$) is the zonal (meridional) wind speed and z is the height.

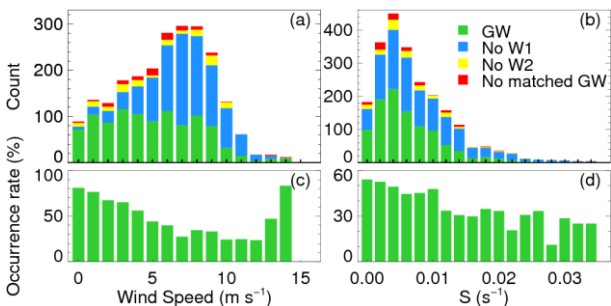

**Figure 6.** The histograms of GWs occurrence with (a) horizontal wind speed and (b) wind shear $S$. The green, blue, yellow and red bars represent the counts of windows with dominant GWs, without potential GWs W1, without W2, without W1 and W2 matched, respectively. W1 and W2 are defined in Appendix B and Fig. B1. GWs occurrence rate versus (c) horizontal wind speed and (d) wind shear $S$.

The histograms of GW occurrence with horizontal wind speed and wind shear in all temporal spatial windows are shown in Fig. 6a and 6b, respectively. The corresponding relationships between GW occurrence rate and horizontal wind speed and wind shear are shown in Fig. 6c and 6d, respectively. It is significant that weaker horizontal wind and weaker wind shear are beneficial to GWs occurrence in ABL from these results. Thus it could be inferred that weak wind and weak wind shear are favorable to the existence of such high frequency wave motions.

The wind rose of the horizontal wind during the field experiment is shown in Fig. 7a. It is apparent that northeasterly wind and southwesterly wind are prevailing around the station in the ABL. It is interesting to note that the long-narrow plain area along Yangtze River around Anqing between Huangshan and Dabie Mountain is also along the direction of northeast-southwest as shown in Fig. 1a. The typical elevations of Huangshan and Dabie Mountain are about 1~2 km. Strong wind along northwest-southeast direction may be blocked in the ABL, thus leading to the weak wind along northwest-southeast direction after the wind flowing over Huangshan or Dabie Mountain and the prevailing wind along northeast-southwest direction. The azimuthal distribution of GW occurrence rate is presented in Fig. 7b. GW occurrence rates are relatively





higher along the northwest-southeast direction than that along the northeast-southwest direction. Interestingly, it seems that the azimuthal distribution of GW occurrence rate is quadrature with the corresponding wind rose. Considering the azimuthal distribution of GW occurrence rate only under weak wind conditions (with horizontal speed less than 4 m s⁻¹), the GW occurrence rates are still relatively higher along the northwest-southeast direction, which is not shown here. The effect of

weak wind on GW occurrence can be excluded here. Hence we can imagine that Dabie Mountain and Huangshan may have an impact on GWs in Anqing. However, surrounding hills around the station as shown in Fig. 1b may also affect the generation and existence of GWs. The effect of surrounding hills will be studied by numerical simulations in next section.

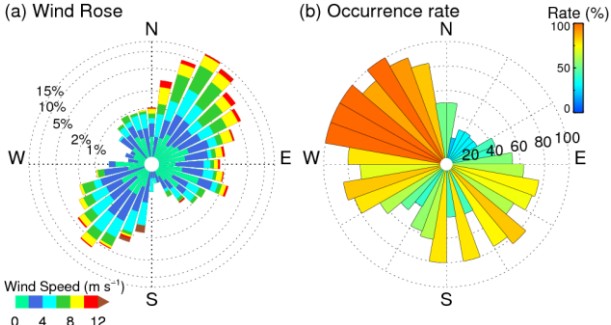

**Figure 7.** (a) Wind rose of horizontal wind in all temporal spatial window during this experiment. It should be noted that the
value of the radius is logarithmic. (b) The azimuthal distribution of GW occurrence rates for every 10 °azimuth angle.

## 4 CFD simulations

Wavelike motions are common in the stably stratified ABL that maybe generated by topography or jet stream (Mahrt, 2014). There is a complex topography around the station as shown in Fig. 1b and a low-level jet in the ABL as shown in Fig. 2. Both of them may be responsible for the generation of the persistent GWs. In order to identify the potential source of the

ducted GWs, a numerical simulation based on CFD is performed to simulate the fluid flow field. Numerical simulations with CFD have better accuracies than mesoscale model for atmospheric flow dynamics in the ABL (Ren et al., 2018). The impact of different boundary conditions, e.g., wind profile and topography, on atmospheric dynamics can be effectively evaluated by changing the boundary conditions. In addition, the numerical simulations can provide the full information of the GWs, which cannot be detected by lidar in this experiment, such as horizontal wavelength, horizontal phase speed and so on.

Therefore, CFD simulations are helpful to investigate GWs in the ABL.

### 4.1 Model description

Reynolds-averaged Navier-Stokes Simulation (RANS) has been widely used to investigate wind field over the past few decades (Toparlar et al., 2017). Compared with Large Eddy Simulation (LES), RANS has advantages of low computational cost and the sufficient accuracy. In this study, a two-equation RANS model based on renormalisation group (RNG) methods

is used to simulate wind filed. The RNG k-ε model was developed to renormalize the Navier-Stokes equations which are account for the effects of smaller scale motions (Yakhot et al., 1992). The model transport equations are obtained as follows:



$$\frac{\partial}{\partial t}(\rho k) + \frac{\partial}{\partial x_i}(\rho k u_i) = \frac{\partial}{\partial x_j}\left(\alpha_k \mu_{eff} \frac{\partial k}{\partial x_j}\right) + G_k + G_b - \rho \varepsilon - Y_M + S_k \tag{2}$$

$$\frac{\partial}{\partial t}(\rho \varepsilon) + \frac{\partial}{\partial x_i}(\rho \varepsilon u_i) = \frac{\partial}{\partial x_j}\left(\alpha_\varepsilon \mu_{eff} \frac{\partial \varepsilon}{\partial x_j}\right) + G_{1\epsilon} \frac{\epsilon}{k}(G_k + C_{3\varepsilon} G_b) - C_{2\epsilon} \rho \frac{\varepsilon^2}{k} - R_\varepsilon + S_\varepsilon \tag{3}$$

where $t$ and $\rho$ are time and air density, $k$ and $\varepsilon$ are turbulence kinetic energy (TKE) and TKE dissipation rate, $x_i$ and $x_j$ are the displacement in dimension $i$ and $j$, $u_i$ is velocity in dimension $i$, $\alpha_k$ and $\alpha_\varepsilon$ are the inverse effective Prandtl numbers for

$k$ and $\varepsilon$, $\mu_{eff}$ is effective viscosity, $G_k$ and $G_b$ represent the generation of TKE due to the mean velocity gradients and buoyancy, $Y_M$ represents the contribution of the fluctuating dilatation in compressible turbulence to the overall dissipation rate, $S_k$ and $S_\varepsilon$ are user-defined source terms, $G_{1\epsilon}$, $C_{2\epsilon}$ and $C_{3\varepsilon}$ are constants.

To simplify the numerical simulation processes, a two-dimensional (2D) rectangle computational domain is applied in this study with 70 km in horizontal and 5 km in vertical from sea level. The upper interface extended to 5 km is set as symmetric

condition to prevent the influence of upper interface on the region concerned that below 2 km. The vertical height of the first layer of grid cells is 5 m. The spatial resolution is approximately 20 m in both horizontal and vertical. The total number of computational grid cells is 875,000. The velocity-inlet is westerly and constant in the west boundary of the computational domain. The easterly interface is set as pressure-outlet boundary to improve reversed flow. The topography is set as no-slip wall condition. The simulation is run with a time step of 0.5 s.

**4.2 Numerical simulations**

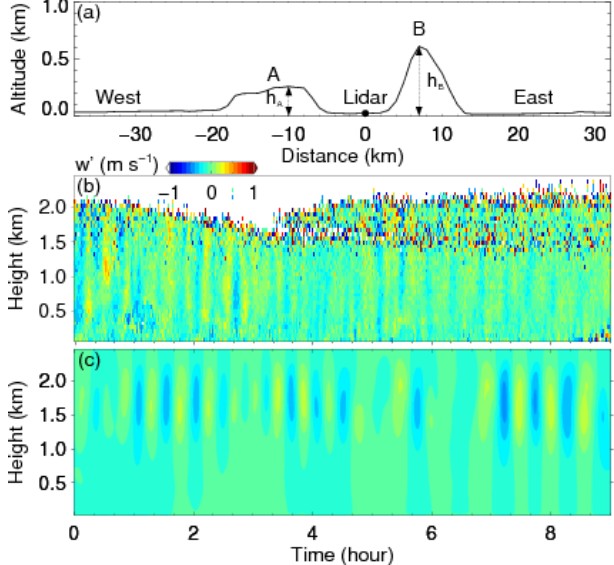

**Figure 8.** (a) The topography used in CFD simulations. A/B represents west/east hill. $h_A$ and $h_B$ represent the height of A and B. (b) The vertical wind perturbation from lidar during 00:00~09:00 LT in 5 September 2018. A mean value is subtracted. (c) The CFD simulated results of vertical wind.





The initial smoothed topography is shown in Fig. 8a. The horizontal location of the domain is roughly along the zonal white dash dotted line in Fig. 1b. This is because the low-level jet and the background wind are mainly in the zonal (east-west) direction. The left (west) hill is defined as A with height of $h_A$, as well as B with height of $h_B$ for right (east) hill in this study. The lidar is located between A and B. The 1-hour mean zonal wind under 2 km from lidar and zonal wind above 2 km

from ERA5 reanalysis data at 00:00 in 5 September 2018 are merged as the sustained import wind profile in the west boundary of the computational domain. The measured vertical wind perturbation $w'$ subtracted by a mean value is shown in Fig. 8b. The vertical wind from CFD simulations is shown in Fig. 8c. It is obvious that a similar wave motion with similar amplitude and period exists in the ABL. This result verifies the accuracy of the CFD numerical simulation results in this study. A small movie of the zonal wind and vertical wind in the whole computational domain can be downloaded from the

Supplement. From this movie, the zonal wavelength can be estimated as ~3 km, and the corresponding zonal phase speed of ~2 m s$^{-1}$. In addition, Kelvin-Helmholtz billows exist in the low-level jet around the altitude of 2 km. These billows may be in relation to these GWs.

Table 2. The wind profile and topography for each case in CFD simulations.

| Case | Wind profile | Topography | Case | Wind profile | Topography |
|------|--------------|------------|------|--------------|------------|
| 1 | $u_z = u_0$ | $h_A \times 1, h_B \times 1$ | 9 | $u_z = u_0$ | $h_A \times 0, h_B \times 0$ |
| 2 | $u_z = 1$ m s$^{-1}$ | $h_A \times 1, h_B \times 1$ | 10 | $u_z = u_0$ | $h_A \times 0, h_B \times 1$ |
| 3 | $u_z = 5$ m s$^{-1}$ | $h_A \times 1, h_B \times 1$ | 11 | $u_z = u_0$ | $h_A \times 1, h_B \times 0$ |
| 4 | $u_z = 10$ m s$^{-1}$ | $h_A \times 1, h_B \times 1$ | 12 | $u_z = u_0$ | $h_A \times 2, h_B \times 0$ |
| 5 | $u_z = u_0 \times 2$ | $h_A \times 1, h_B \times 1$ | 13 | $u_z = u_0$ | $h_A \times 4, h_B \times 0$ |
| 6 | $u_z = u_0 \times 4$ | $h_A \times 1, h_B \times 1$ | 14 | $u_z = u_0$ | $h_A \times 6, h_B \times 0$ |
| 7 | $u_z = u_0 + 5$ m s$^{-1}$ | $h_A \times 1, h_B \times 1$ | 15 | $u_z = u_0$ | $h_A \times 0, h_B \times 2$ |
| 8 | $u_z = u_0 + 10$ m s$^{-1}$ | $h_A \times 1, h_B \times 1$ | 16 | $u_z = u_0$ | $h_A \times 0, h_B \times 4$ |

Based on this result, wind profiles $u_z$ with different wind shear and topography with different height of hills A and B are

employed in the CFD numerical simulations. A detailed boundary conditions are listed in Table 2. The merged zonal wind at 00:00 in 5 September 2018 used in Fig. 8 is defined as $u_0$. The corresponding simulated results of zonal wind and vertical wind above the lidar for all cases are shown in Fig. 9, respectively. It should be noted that the flow solution is initiated as a steady state in all cases except case 7 and 8. In case 7 and case 8, the start time of 0 is defined as when the simulations started running and the velocity-inlet flowed from the west boundary at the same time.







**Figure 9.** The simulated zonal wind u and vertical wind w above the lidar for all 16 cases as described in Table 2.

In case 1, persistent wave motions are not only exist in vertical wind, but also in zonal wind near and below the low-level jet around 2 km as shown in Fig. 9. It is consistent with lidar detections as shown in Fig. 4. It is obvious that no wave motions

5    generated with uniform wind speed of 1, 5 and 10 m s$^{-1}$ in case 2~4. Thus GWs cannot be excited without wind shear here.



From the results of case 1, 5 and 6, the wave amplitudes and frequencies increase with the enhancement of wind shears. For case 1, 7 and 8, no persistent wave motions exist with the wind speeds increasing without enhancement of wind shears. Only several solitary wavelike motions can be found when the wind flow passed by the lidar and dissipated rapidly. In addition, the wave motions in case 1~8 are mainly exist under 2.5 km where wind speeds are relatively weak. Therefore, it can be

inferred that it is beneficial to the generation of persistent waves under persistent weak wind conditions, which is consistent with previous result in Sect. 3.2.

What will happen when the height of hills A and B near the lidar location are changed? In case 9~11, persistent wave structures still exist with only a few changes when the hills A and/or B disappeared. From case 9 and 11~13, persistent wave structures always exist and do not change significantly. When the height of A increased to $h_A \times 6$ in case 14, i.e., the height of

the low-level jet near 2 km altitude, the zonal wind structure changes significantly. In case 9~10 and 15~16, the wave motions also exist and do not change significantly even though the height of B increased to the height of the low-level jet, $h_B \times 4$.

Therefore, based on these results of simulation cases, persistent GWs are excited by the persistent wind shear around the low-level jet. The wave structures mainly occur under weak winds. The topography, i.e., the around hills near the station as

shown in Fig. 1b, plays a negligible role in the GW generation. Nevertheless, the topography may play a more important role in the downstream when the height of jet is comparable with the height of topography.

## 5 Discussion

Based on the above experiments and simulations, the mechanism of the persistent wave motions can be inferred as follows. A westerly low-level jet of ~5 m s$^{-1}$ exists above the background southeasterly light wind. The light wind may be in relation

to Huangshan and Dabie Mountain. The weak wind shear around the low-level jet may lead to the appearance of wave motions in the light wind. In addition, a strongly stable thermal stratified ABL with inversion layer occurs during the night in Anqing. Negative values of $N^2$ appear near the altitude of ~2 km. Thereby, the wave motions may be trapped in a ducted structure with long lifetime. The GWs exist without apparent phase progression with altitude in the whole ABL from the surface to the height of ~2 km.

Such quasi monochromatic waves with multi wave cycles and approximately constant period and amplitude are infrequently observed in the ABL (Mahrt, 2014). Nevertheless, similar quasi monochromatic wave motions with multi wave cycles have been reported in several studies. Banakh and Smalikho (2016) revealed a coastal-mountain lee wave with period of ~9 min at daytime on 23 August 2015 in the stable stratified ABL on the coast of Lake Baikal. The wave exists between 100 m and 900 m height range with a lifetime of about 4 h. This wave was suggested to be in relation to the presence of two narrow jet

streams at heights of about 200 m and 700 m above ground level. Similar wave motions were also detected in the vertical wind accompanied with a low-level jet in Banakh and Smalikho (2018). It is regrettable that the authors have not given a discussion on the contaminated wavelike motions from 01:00 to 08:00, except the internal wave with period of ~6 min at





07:00. Fritts et al. (2003) reported wave motions with periods typically 4~5 min below the height of ~800 m under light wind with a low-level jet, clear sky conditions throughout the night of 14 October 1999. These wave motions were interpreted as ducted GWs that propagate horizontally along maxima of the stratification and mean wind, and that are evanescent above, and possibly below and/or between, the ducting level(s) (Fritts et al., 2003). Viana et al. (2009) also reported a ducted

mesoscale gravity wave over a weakly-stratified nocturnal ABL. This wave lasted less than 10 wave cycles, about 2 hours, with periods of ~16 min. Román-Cascón et al. (2015) analysed non-local GWs generated by wind shear or low-level jet trapped within the stable ABL. With acoustic echo sounder, similar wave motions were also observed without apparent phase progression with altitude in stably stratified ABL several decades ago (Beran et al., 1973; Hooke and Jones, 1986). The wave motions mentioned above were mainly occurred in the stable boundary layer under the height of ~1000 m or even

~100 m, while the wave motions exist from surface layer to as high as ~2000 m in our study due to different capability of measurements. In a previous study presented by Wang et al. (2019), obvious wave motions with periods of 10~30 min in vertical wind were observed in the whole residual layer from 1 June 2018 to 2 June 2018 by a similar CDL system.

The mechanism of this long-live GWs is in consistent with other similar wave motions referenced above in some aspects. Low-level jet or wind shear is one of the mainly sources of such wavelike motions in the ABL. Stably stratified ABL usually

leads to effective ducting quasi monochromatic wave motions with a long lifetime and multi wave cycles. Although wind shear near the low-level jet is the main source of GWs as discussed in Sect. 4.2, weaker wind shear is more favorable to the existence of high frequency GWs. Similarly, the surrounding hills play negligible roles in generating the ducted GWs while Dabie Mountain and Huangshan may have an impact on the generation and existence of GWs.

The vertical structure of GWs is describe by Taylor-Goldstein equation (Gossard and Hooke, 1975):

$$m^2 = \frac{N^2}{c_i^2} + \frac{\bar{u}_{zz}}{c_i} - k_h^2 - \frac{1}{4H^2} \qquad (4)$$

where $m$ is the vertical wavenumber, $c_i$ is the intrinsic phase speed in the direction of propagation, $\bar{u}_{zz}$ is the second derivative with height of the mean wind in the direction of wave propagation, $k_h$ is the horizontal wavenumber, $H$ is the scale height. A sufficiently deep atmospheric layer is required for a wave duct with positive values of $m^2$. To resolve this equation, the vertical profile of squared buoyancy frequency, which is calculated by temperature profile measured by

radiosonde, is shown in Fig. 5b. Simultaneous hourly mean wind, required to resolve $c_i$ and $\bar{u}_{zz}$, can be obtained from lidar measurements. However, the horizontal structures of this wave motion, i.e., $c_i$ and $k_h$, are still unclear in this study. Horizontal structures of wavelike motions in ABL can be detected by airborne lidar (Chouza et al., 2016; Witschas et al., 2017) and ground-based lidar with range height indicator (RHI) scans (Poulos et al., 2002; Wang, 2013) or plan position indicator (PPI) scans (Mayor, 2017). GW parameters, such as horizontal phase speed, horizontal wavelength, propagation

direction, intrinsic frequency, can be resolved from these measurements. Nevertheless, we try to illustrate the character of this ducted wave for a plausible propagation direction and horizontal wavelength from CFD simulations. The propagation direction is assumed to be westerly here as the simulated wave is westerly from the movie in the Supplement. Thus the horizontal wavelength is equal to zonal wavelength which is estimated as ~3 km in Sect. 4.2.



The vertical profile of vertical wave number squared is shown in Fig. 10. The singular point of the relative maxima $m^2$ in right panel is caused by critical level where the intrinsic frequency is Doppler-shifted close to zero. A ducting process occurs between ~1.5 km altitude and the ground where $m^2 > 0$. It is a result of the combination of thermal and Doppler ducts. The thermal duct is dominant under the temperature inversion with maxima buoyancy frequency squared for all propagation

directions as shown in Fig. 5. The Doppler duct is dominant between ~0.5 and ~1.5 km altitude range due to the critical level induced by the low-level jet of wind maximum in a particular direction. Thus the ducted motions give a plausible explanation for the trap of the long-live GW in the ABL.

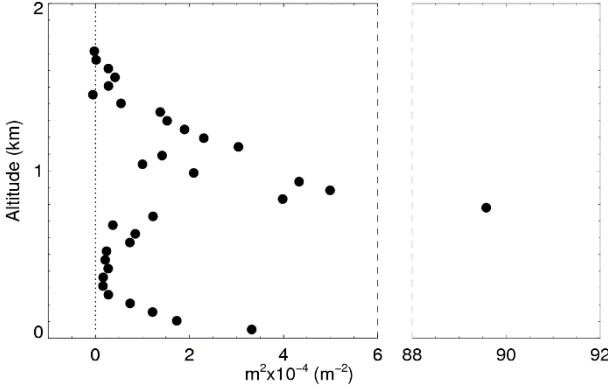

**Figure 10.** The vertical profile of vertical wave number squared. The dotted line represents zero line.

## 6 Conclusion

A persistent wave motion was investigated by experiments and numerical simulations. From 4 to 5 in September 2018, GWs with periods of 10~30 min were observed in the whole ABL from ground to the height of ~2 km by a coherent Doppler lidar during a field experiment in Anqing. The amplitudes of these GWs were about 0.15~0.2 m s$^{-1}$ in vertical wind. These GWs existed more than 20 wave cycles. The periods were about 15~25 min before 03:00 LT and 20~30 min after that. A westerly low-level jet was observed at the altitude of 1~2 km in the ABL with maxima speeds of 5~10 m s$^{-1}$. Simultaneous

temperature profiles from radiosonde measurement and ERA5 reanalysis data confirmed the existence of a strong stably stratified ABL. There was an inversion layer under the altitude of ~500 m and negative buoyancy frequency squared near the height of ~2 km. Note that there was no apparent phase progression with altitude of these GWs. Moreover, the vertical and zonal perturbations of the GWs were apparent quadrature with vertical perturbations generally leading zonal perturbations. These characteristics suggested that such GW motions are ducted GWs trapped in the ABL which is also verified by the

vertical structure of the wave motions. In addition, the relations between high frequency GWs and background wind conditions were also analysed. High frequency GWs were likely to exist under weak wind and weak wind shear. When the horizontal wind was along the northwest-southeast direction, where Huangshan and Dabie Mountain located, the occurrence rate of such GWs was higher in the ABL. Based on simplified 2-D CFD numerical simulations, the generation mechanisms of such GWs were discussed. The low-level jet streams were considered to be responsible for the excitation of GW motions





in present study. Wave motions mainly occurred under weaker wind conditions, which is consistent with previous result and other studies of such ducted waves. The contributions from flow over the surrounding hills could be ignored.

The current study contributes to our understanding of GWs generation mechanism in the ABL, which plays a key role in atmospheric dynamics. Furthermore, the National Meteorological Observing Station of Anqing is close to an airport as shown in Fig. 1b, which will be affected by the clear air turbulence caused by breaking GWs. The application of such coherent Doppler lidar will enhance the measurement capability with high quality data in the ABL, thus enriching our knowledge and improving our abilities in aviation safety, weather forecast and climate models in future. However, the horizontal structures of the GWs are still unclear in this study. Simultaneous measurements with multi lidars and multi scanning modes are required in further studies.

10  *Data availability*

The ERA5 data sets are publicly available from ECMWF website at https://www.ecmwf.int/en/forecasts/datasets/reanalysis-datasets/era5, last access: 1 March 2019. The elevation data are available at SRTM website (http://srtm.csi.cgiar.org, last access: 1 March 2019). Lidar and radiosonde data can be downloaded from http://www.lidar.cn/datashare/Jia_et_al_2019.rar, last access: 16 March 2019.

15  **Appendix A: Lidar observational results during the experiment**

The vertical wind, horizontal wind speed, wind direction and CNR during the field experiment from 16 August to 5 September 2018 are shown in Fig. A1~A4, respectively.



**Figure A1.** Time height cross section of vertical wind speed per day during the experiment. Dates are shown in the top left of each panel, and are read as YYYY/MM/DD.







**Figure A2.** Similar to Fig. A1 but for horizontal wind speed.

**Figure A3.** Similar to Fig. A1 but for horizontal wind direction.





**Figure A4.** Similar to Fig. A1 but for CNR.

## Appendix B: Identification of dominant high frequency GWs

In order to extract high frequency GWs with periods of 5~50 min, a window of 4-hour length and 200-m height shifted in
5    steps of 1 hour temporally and 100 m vertically is used. First, height averaged temporal profiles are obtained in a given





window, such as vertical wind in Fig. B1a and B1b. Second, the corresponding Morlet wavelet power spectrum and Lomb–Scargle (LS) periodogram are calculated as shown in Fig. B1b~1f. Significance levels of LS and wavelet are used to identify GWs. Period ranges of 5~50 min are indicated by the vertical dotted lines in Fig. B1e and B1f. Thirdly, maxima values in LS spectral larger than significance level within periods of 5~50 min are identified as the potential waves W1 as shown in Fig.

B1e and B1f. Then, the average of wavelet significance in the central 1-hour areas, between the skyblue dashed lines in Fig. B1c and B1d, are plotted as the brown dash dotted lines as shown in Fig. B1e and B1f. Potential waves W2 are also identified by the significance levels. Following this, the potential waves W1 and W2 are compared. W1 and W2 with maximum amplitudes and relative differences in periods less than 30% are considered to be the best matched pairs. For example, W1 and W2 are well matched in Fig. B1e while are not matched in Fig. B1f. Finally, these matched pairs of

potential waves are identified as the dominant GWs in a specified window. If no W1 or W2 are recognized, or no pairs of W1 and W2 are matched, it skips to the next temporal-spatial window. Thus we can know whether there is a dominant high frequency wave motion in a specified temporal-spatial window or not.

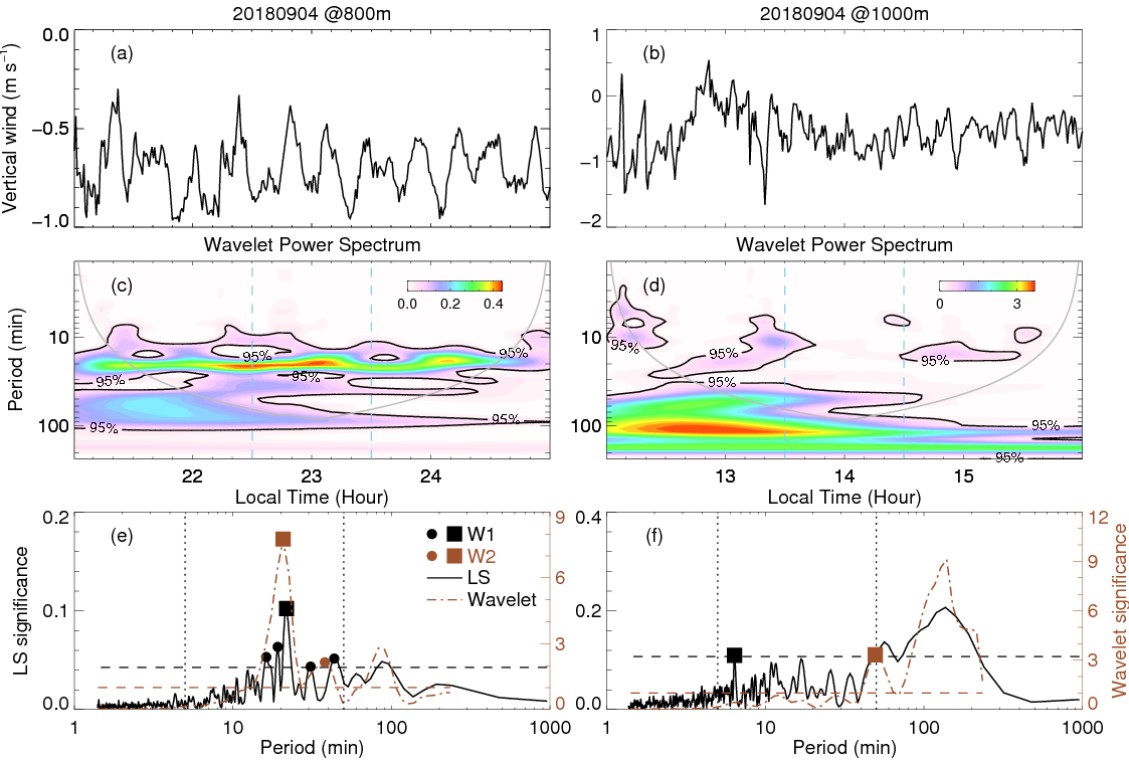

**Figure B1.** (a) Average temporal vertical wind in a given window. (c) Morlet wavelet power spectrum of vertical wind. The
black contours indicate significance level of 95%. The gray line represent the Corn-of-Influence (COI). The skyblue vertical dashed lines indicate the central one hour area of the temporal window. (e) LS periodograms of the vertical wind (black solid line). Average signicance between the skyblue dashed lines in (c) (brown dash dotted line). The black vertical dotted lines represent period range of 5~50 min as high frequency waves. Black and brown dashed lines indicate the significance levels of LS periodograms and wavelet, respectively. The black solid circles represent potential waves W1, while the brown represent
potential W2. Solid squares indicate the identified waves. (b)(d)(f) are same as (a)(c)(e) but in a different temporal-spatial window.



However, we note that such a simple and rough method may result in unrealistic wave signals due to the false detection. During the START08 field experiment, Zhang et al. (2015) confirmed that part of the wavelike motions may be due to intrinsic observational errors or other physical phenomena (e.g., nonlinear dynamics, shear instability and/or turbulence). Nevertheless, the relation between high frequency GWs and weak wind is consistent with CFD numerical simulations in Sect. 5  4 and previous studies referenced in Sect. 5. Thus the methodology of dominant GW identification is still reasonable in this manuscript.

*Author contribution*

HX conceived, designed the study. YW, LZ CW and MJ performed the lidar experiments. MJ, CW and TW performed the lidar data analysis. LL, DL and RC provide the field experiment site and the radiosonde data. JW analyzed ERA5 data. JY 10  performed the CFD numerical simulations. MJ and JY carried out the analysis and prepared the figures, with comments from other co-authors. MJ, HX, XX and XD interpreted the data. MJ, JY and HX wrote the manuscript. All authors read and approved the final manuscript.

*Competing interests.*

The authors declare that they have no conflict of interest.

15  *Acknowledgements.*

We acknowledge the use of ERA5 data sets from ECMWF website at https://www.ecmwf.int/en/forecasts/datasets/reanalysis-datasets/era5. We acknowledge the use of elevation data sets from SRTM website at http://srtm.csi.cgiar.org.

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
