# Peer review of "Long-lived high frequency gravity waves in the atmospheric boundary layer: observations and simulations"

_Atmospheric Chemistry and Physics, 2019_

## Referee Comment (RC1) · Anonymous Referee #1 · 26 Jun 2019

**Summary:**

This manuscript presents observations of persistent gravity wave activity in a stable atmospheric boundary taken as part of a field experiment over Anqinq, China. Alongside the observations, a series of CFD simulations are conducted in order to understand the likely mechanisms for generation of these gravity waves. The results are not particularly novel – gravity waves have been observed in previous studies of the ABL and shear is known to be an important mechanism for generation of gravity waves. Having said that, the observations make a nice case study, bringing together Doppler lidar observations with some radiosonde data to give an unusually detailed set of observations

of such waves.

**Major comments:**

1) The general motivation for the work is perhaps a little misleading. The introduction talks about the breaking of gravity waves at critical layers at altitude and the role of high frequency waves in momentum transport into the upper troposphere. The observations presented are of vertically coherent waves which look much more like horizontally propagating trapped wave modes to me, and hence it is not clear the relevance of either of these? Such horizontal modes might well be important and linked to low level turbulence (e.g. the role of trapped waves in generation of rotors as in the T-Rex experiment, Grubisic et al, 2008) and they have also been linked to the initiation of convection (e.g. Lac et al, 2002; Marsham and Parker, 2006; Birch et al, 2012). You might also mention low level wave drag (e.g. Lapworth and Osborne, 2016; Tsiringakis et al, 2017). The introduction also suggests (page 1, line 29) that waves are only generated in the troposphere, which is not true.

It is also not clear what the specific motivation for this work is. How might this work help to i) deepen our understanding of the mechanism leading to the generation and propagation of gravity waves in the boundary layer and/or ii) improve our capability to model these waves or parametrise their effects? How might the results be useful to researchers in other regions of the world? You probably want to return to these questions in the conclusions and discussion.

2) It would be useful to the reader, and increase the impact of your work, if you can highlight the novelty of this work given that various other studies have observed and modelled gravity waves in the boundary layer, including many of the papers referenced in the introduction.

3) Table 1 gives the spatial resolution. I assume this is the along-beam range gate size? It would be useful to clarify this. There are two additional factors to consider. Firstly, the beam spread angle will alter the effective horizontal resolution of each sample

(increasing with height). Perhaps more importantly the Doppler wind retrieval requires multiple scans (1 vertical and at least 2 at an angle – here 30°). This means the wind retrieval is over a cone with a much larger width than the 60m resolution given. At 2km the radius of this cone would be 1150m. This might be a significant fraction of the wavelength of a short, high frequency wave. These issues should at least be discussed.

4) Bottom of page 4. These jets are important as the shear associated with them is hypothesised to lead to the gravity waves. It is not clear what the cause of these jets is though. The big changes in wind direction are unusual. Can you offer any explanation? It is possible that the upper jet is some sort of nocturnal low level jet as it seems to strengthen during the night and weaken a little around dawn, but there is not much evidence of wind turning with time as one might expect to see if this was an inertial oscillation. Have you tried looking at hodographs at a specific height from the lidar winds to see evidence of this? Are there other explanations, e.g. the height of the surrounding topography leading to channelling in the valley?

5) Page 6, line 2. It seems likely that this aerosol concentration profile is associated with a stably stratified boundary layer, but I'm not sure that you can definitively say this from the lidar observations since you have no direct measure of temperature or stability.

6) Page 7, figure 5. It would be useful to include a plot of the Scorer parameter here in addition to $N^2$ since it is hypothesised that wind shear is important.

7) Page 7, lines 17-18. I am not sure you can say anything about the presence of absence of waves above 1800-2000m. The lidar barely extends above 2000m between 0400-0900 and the CNR is quite low, with noisy $w$ fields. I would reword this conclusion.

8) Page 8, Figure 6. If you are plotting W1 and W2 in the text, then you need to at least briefly explain them here. The main text should be understandable without reading the appendix. The appendix is for additional details. At the moment the figure

is meaningless without knowing what it means having only W1 or only W2. (Actually this is still not very clear, even after reading the appendix – see below).

9) Page 8, Figure 6. Also, are these plots are for the whole field campaign or just the case study on 4th - 5th Sept. What height did you go up to? How was this chosen?

10) Page 9, lines 15-16. This is a very bold claim! Both the terms "mesoscale models" and "CFD models" can cover a wide range of different things. Ultimately both are solving the same equations. Which is better will depend on the details of individual models and their numerics, and the setup of individual simulations (e.g. resolution, turbulence scheme etc. . ..). CFD models traditionally do not include many atmospheric processes, although the distinction is increasingly blurred. All your advantages could equally apply to a high resolution atmospheric model. I would just remove this sentence altogether.

11) Page 9, section 4.1. So which CFD model are you actually using? Is this a commercial code? Include a reference to the actual model and its validation if at all possible.

12) Page 10, lines 13-14. With the first cell height at 5m then you are not resolving the roughness sub-layer at all, and so you need to apply some sort of wall function / Monin-Obukov similarity function rather than just the no-slip boundary condition. Are you doing this? If so, what?

13) Presumably there is a prognostic equation for potential temperature or similar, in order to include stability effects? No mention of this. What boundary conditions are used for this variable?

14) Page 11, line 9. What do you mean by "symmetric condition" at the upper interface? Which variables does this apply to?

15) Page 11, lines 17-18. I don't understand the comment that the flow solution is initiated as a steady state in all cases except case 7 and 8. How can you initiate at a steady state? Do you mean you initialise with the merged wind profile everywhere? In

general this will not be an exact solution to the model equations, even in 1-d, so this isn't a steady state. Even if it was a solution in 1-d, the inclusion of topography means there will be variations across the domain which you cannot know without solving first. I assume the additional complication for cases 7 and 8 is that the addition of a constant velocity everywhere would break the no-slip boundary condition. How do you deal with that? Is the velocity near the surface reduced to produce a consistent initialisation / inlet profile? If so, how?

16) Page 11, table 2. You don't actually define $u_0$. I assume this is the merged wind profile (lines 4-6), in which case define it there. It would be useful to have a plot of this model wind profile, and also the potential temperature profile.

17) The simulations presented are all 2-D, however the analysis elsewhere suggests that topography might be constraining the low level flow. Why choose to conduct 2-D simulations? What is the impact of this? This choice needs to be justified, and the limitations discussed.

18) Page 13, lines 15-16. If these waves were topographically generated, one would expect stationary waves rather than the propagating waves seen here. You have only shown time-height plots. If you look at vertical cross sections is there evidence of stationary topographically generated gravity waves at all?

19) While there are a relatively comprehensive set of simulations presented here, there is relatively little attempt to explain physically why the differences between the simulations occur. It might be fruitful to look at the Scorer parameter for different wind profiles. Does this explain the differences in wave trapping observed for example? One factor which is not investigated at all is the role of stability. Without stability there would be no waves at all, and the presence of these trapped horizontal wave modes must be at least partly due to the low level inversion. How would altering this affect the results?

20) Page 13, line 22. It is rather unusual to see negative values of $N$ at 2km. This implies an unstable atmosphere at this height. Do you have any idea what is driving

this? I note that these negative values are fairly small, and only over a narrow height range (Figure 5b). It is interesting to note that the waves appear to reach well above the stable boundary layer in this case (perhaps 200-300m depth), even though there are several near-neutral levels below 2km. Can you explain this? Plotting temperature profiles (rather than potential temperature) and the choice of scales for the $N^2$ plot makes this difficult to judge though.

21) Appendix B. From what is written I cannot see why you need to choose and compare two separate wave frequencies W1 and W2. From figure B1 I am guessing that W1 is the most significant peak with LS and W2 the most significant peak with the Morlet wavelet? The text does not explicitly say this. Similar, I am assuming from the figure that the wave is identified as the most significant peak, although this is not explicitly stated. Only when similar waves are identified by both methods is the case treat as a GW. Why do you use this criterion? This whole section could be better explained.

22) Page 22, line 1. Do you have any evidence of false detection? Are any of the wave signals unrealistic? There doesn't seem much evidence to confirm or deny this at the moment so it is rather speculative.

**References**

Birch et al (2013) Impact of soil moisture and convectively generated waves on the initiation of a West African mesoscale convective system. QJRMS 139 1712-1730. Grubisic et al (2008) THE TERRAIN-INDUCED ROTOR EXPERIMENT: A Field Campaign Overview Including Observational Highlights. BAMS 89 1513–1534. Lac, Lafore and Redelsperger (2002) Role of Gravity Waves in Triggering Deep Convection during TOGA COARE. JAS 59 1293–1316. Lapworth and Osborne (2016) Evidence for gravity wave drag in the boundary layer of a numerical forecast model: A comparison with observations. QJRMS 142 3257–3264. Marsham and Parker (2006) Secondary initiation of multiple bands of cumulonimbus over southern Britain. II: Dynamics of secondary initiation. QJRMS 132 1053-1072. Tsiringakis, Steenveld, Holtsag (2017)

Small-scale orographic gravity wave drag in stable boundary layers and its impact on synoptic systems and near-surface meteorology. QJRMS 143 1504–1516.

**Minor comments:**

Title. "Long-lived High Frequency Gravity Waves in the Atmospheric Boundary Layer" would be better English.

Page 4, line 4. "Wind field"

Page 4, line 20. "4-dimensions"

Page 6, Fig 3 caption. Should this be "Cone-of-Influence"?

Page 7, line 15. Units should not be in italics.

Page 9, line 2. The word "quadrature" is not really appropriate here. I would say "is perpendicular to the corresponding wind rose". This occurs at other places in the text too.

Page 9, line 25. "simulate the wind field".

Page 12, figure 9 caption. "wave motions are not only exist in the vertical wind". "no wave motions are generated". "cases 2-4"

Page 15, line 18. "... perturbations of the GW were $90°$ out of phase with vertical perturbations..."

---

## Referee Comment (RC2) · Anonymous Referee #2 · 13 Aug 2019

Comment on the manuscript "Long-live High Frequency Gravity Waves in Atmospheric Boundary Layer: Observations and Simulations" by Mingjiao Jia et al.

The manuscript presents the results of lidar observations of wave like variations of wind velocity vector vertical and horizontal components in the boundary layer of atmosphere obtained during the field experiment in August-September 2018 in the location of Anqing, China. The experimental results are accompanied by the results of model numerical simulation of the wind field disturbed by the topographical objects. Based on the obtained experimental and computational results the conclusions about mechanism of generation of wave variations of wind velocity are formulated in the manuscript.

[Figure]

Major comments: 1) The general goal of the study is not clear from the manuscript. It may be proposed that this goal is study of atmospheric waves arising under conditions of stable thermal stratification in the boundary layer of atmosphere. But only one event of atmospheric wave on 4-5 September is analyzed in the manuscript with the use of information about the profile of temperature in height. Moreover, even for that event there is no data on temperature profile measured with the radiozonde at 19:15 on 4th September in the manuscript. To improve understanding of this issue, it may be useful to present the temperature profiles in height during all the period of field experiment and carry out the analysis of frequency of wave events not only with taking into account the magnitudes of wind velocity and wind shear, as it is presented in Fig. 6, but also with consideration of the temperature stratification. 2) The representativeness of the estimates of the mean wind velocity. As mentioned in line 6 on p.4, measurement duration of radial velocity in one direction is 10 s during this experiment. For used in the experiment scanning geometry such duration of measurements is insufficient in order to obtain statistically justified estimates of the mean wind velocity components. Actually, it is well known that integral spatial scale of wind turbulence is proportional to the height under ground in the lower atmosphere and can reach a few hundreds of meters at the heights 600-2000 m. To obtain statistically justified estimate of the mean velocity, the velocity fluctuations caused by the turbulent inhomogeneities of all the scales up to hundreds of meters must be averaged. Even for observed in the experiment maximal velocity 10 m/s in order to average velocity fluctuations caused by the turbulent inhomogeneities of velocity field of such spatial scales it requires few hundreds seconds, at least. 3) What is the reason of variations of wave period in Figs 3, 4? Model calculations in Fig. 8 do not reproduce wave period variations. It may be useful to compare the experimental and calculation results in more detail by combining the experimental and calculated data in one plot. It is difficult to compare and understand the results in Figs. 8b, 8c. 4) The code used for numerical modeling must be described in more detail. As it can be proposed, some version of the program CFD Fluent was used in the modeling. Accordingly to Eqs. (2), (3), it is required to set a lot of input

turbulent parameters in order to perform the modeling using that code. None of these input parameters is determined experimentally. At least, there is no information about that in the manuscript. If so, there is no any base for quantitative comparison of the experimental and computational results and conclusions about the mechanism of wave generation. Minor comments: 1) Temperature profile curves in Fig.5 should be identified. 2) Parameter N in line 14, p.7 should be expressed by formula. 3) Resolution of wind and temperature experimental data in height should be indicated. 4) Magnitudes of âĐŐA, and âĐŐB in Fig.8 and Table 2 must be indicated.

---

## Author Comment (AC1) · 30 Sep 2019

We appreciate all the great efforts and constructive comments from the reviewers. We have revised the manuscript carefully according to the reviewers' comments and suggestions. Our point-by-point responses and changes are appended below in blue fonts.

**Anonymous Referee #1**

**Summary:**
This manuscript presents observations of persistent gravity wave activity in a stable atmospheric boundary taken as part of a field experiment over Anqinq, China. Alongside the observations, a series of CFD simulations are conducted in order to understand the likely mechanisms for generation of these gravity waves. The results are not particularly novel – gravity waves have been observed in previous studies of the ABL and shear is known to be an important mechanism for generation of gravity waves. Having said that, the observations make a nice case study, bringing together Doppler lidar observations with some radiosonde data to give an unusually detailed set of observations of such waves.
**Response**: Thanks a lot for your comments. Although GWs in ABL have been observed in previous studies and shear is known to be an important mechanism. GWs from surface to ~2 km with long lifetime more than 10 hours and 20 wave cycles are reported rarely. Both observations and simulations are used to analyze this unique GW activity. Therefore, we believe that this study of long-lived GWs in ABL is novel and interesting to the readers.

**Major comments:**
1) The general motivation for the work is perhaps a little misleading. The introduction talks about the breaking of gravity waves at critical layers at altitude and the role of high frequency waves in momentum transport into the upper troposphere. The observations presented are of vertically coherent waves which look much more like horizontally propagating trapped wave modes to me, and hence it is not clear the relevance of either of these? Such horizontal modes might well be important and linked to low level turbulence (e.g. the role of trapped waves in generation of rotors as in the T-Rex experiment, Grubisic et al, 2008) and they have also been linked to the initiation of convection (e.g. Lac et al, 2002; Marsham and Parker, 2006; Birch et al, 2012). You might also mention low level wave drag (e.g. Lapworth and Osborne, 2016; Tsiringakis et al, 2017).
**Response**: Thanks for this suggestion. The introduction is rewritten according to your suggestion to emphasize the motivation of this study.
**Changes**: Page 1, line 31 to page 2, line 8. "In general, most of these GWs will propagate upward into the upper atmosphere, e.g. upper troposphere, stratosphere, mesosphere and even thermosphere. Leading to transportation of energy and momentum from lower atmosphere to upper atmosphere, and thus affect the coupling between lower atmosphere and upper atmosphere, as well as dynamic and thermal

structure of the global atmosphere (Fritts and Alexander, 2003; Holton and Alexander, 2000). However, trapped GWs, e.g. trapped lee waves and ducted motions with high frequency and coherent variability, could only propagate horizontally. In the lower atmosphere, these horizontally propagating GWs may be linked to low level turbulence (e.g. rotors), the initiation of convection and low level wave drag (Birch et al., 2013; Grubišić et al., 2008; Lac et al., 2002; Lapworth and Osborne, 2016; Marsham and Parker, 2006; Tsiringakis et al., 2017). Therefore, such trapped GWs play a key role in weather forecast, climate models and aviation safety."

The introduction also suggests (page 1, line 29) that waves are only generated in the troposphere, which is not true.

**Response**: The GWs are usually generated in the troposphere in the introduction (page 1, line 29), not only generated in the troposphere.

It is also not clear what the specific motivation for this work is. How might this work help to i) deepen our understanding of the mechanism leading to the generation and propagation of gravity waves in the boundary layer and/or ii) improve our capability to model these waves or parametrise their effects? How might the results be useful to researchers in other regions of the world? You probably want to return to these questions in the conclusions and discussion.

**Response**: In the 3$^{rd}$ paragraph of Introduction, we have mentioned that most of the observed ducted waves in the ABL are limited to the surface layer within tens or hundreds of meter near ground in previous studies. GWs from surface to ~2 km with long lifetime more than 10 hours and 20 wave cycles are rarely reported. Compared to previous studies, this work will make contribution to deepen and enhance our understanding of such unique long-lived ducted GWs.

**Changes**: Page 14, line 25 to line 31. "The wave motions mentioned above were mainly observed in the stable boundary layer under the height of ~1000 m or even ~100 m, while the wave motions exist from surface layer to as high as ~2000 m in our study due to different capability of measurements. In addition, the lifetime of the ducted GWs is more than 10 hours and 20 wave cycles, while in previous studies listed above, most of the lifetimes are less than hours with several wave cycles. These characters make this ducted GWs unique and novel. However, in one of our previous study, obvious wave motions with periods of 10~30 min in vertical wind were observed in the whole residual layer from 1 June 2018 to 2 June 2018 by a similar CDL system (Wang et al., 2019)."

2) It would be useful to the reader, and increase the impact of your work, if you can highlight the novelty of this work given that various other studies have observed and modelled gravity waves in the boundary layer, including many of the papers referenced in the introduction.

**Response**: Thanks for this suggestion. To the best of our knowledge, other reported observations of such ducted GWs are generally trapped within tens or hundreds of meter near ground and short time over few hours. We have mentioned this in the Introduction and Discussion. To study the GWs, we built a robust and mobile coherent Doppler wind lidar with good detection capability. In the field experiment, continuous wind observation of the GWs are captured to a height about 2000 m over a lifetime of tens

hours. This rarely observed case provides us an opportunity to study such GWs.

**Changes**: Page 14, line 25 to line 31. "The wave motions mentioned above were mainly observed in the stable boundary layer under the height of ~1000 m or even ~100 m, while the wave motions exist from surface layer to as high as ~2000 m in our study due to different capability of measurements. In addition, the lifetime of the ducted GWs is more than 10 hours and 20 wave cycles, while in previous studies listed above, most of the lifetimes are less than hours with several wave cycles. These characters make this ducted GWs unique and novel. However, in one of our previous study, obvious wave motions with periods of 10~30 min in vertical wind were observed in the whole residual layer from 1 June 2018 to 2 June 2018 by a similar CDL system (Wang et al., 2019)."

3) Table 1 gives the spatial resolution. I assume this is the along-beam range gate size? It would be useful to clarify this. There are two additional factors to consider. Firstly, the beam spread angle will alter the effective horizontal resolution of each sample (increasing with height). Perhaps more importantly the Doppler wind retrieval requires multiple scans (1 vertical and at least 2 at an angle – here 30°). This means the wind retrieval is over a cone with a much larger width than the 60m resolution given. At 2km the radius of this cone would be 1150m. This might be a significant fraction of the wavelength of a short, high frequency wave. These issues should at least be discussed.

**Response**: Thanks for this suggestion. These issues are added in Discussions in the revised manuscript. The retrieval of the horizontal wind is based on the hypothesis of a homogeneity wind field on a horizontal plane. Accompanying with wave activities, the radius of the cone will lead to some bias on the horizontal wind if the radius is equivalently to or larger than the scale of the waves. Nevertheless, the effect on the period of the wave motions is negligible. If the radius is smaller than the scale of waves significantly, these bias can be also ignored.

**Changes**: Page 16, line 3 to line 10. "It should be noted that the retrieval of horizontal wind is based on the hypothesis of a homogeneity wind field on a horizontal plane. Accompanying with wave activities, the radius of scanning beams cone will lead to bias on the retrieved horizontal wind. If the radius is equivalently to or larger than the scale of the horizontal wavelength of the GWs, these bias may be significantly affect the result in horizontal component, especially the amplitude of the retrieved GWs. Nevertheless, the bias in the period of the wave motions is negligible. If the radius is smaller than the scale of the horizontal wavelength of the GWs, the bias in both amplitude and period can be ignored. In this study, the horizontal wavelength is estimated as ~3 km in Sect. 4.2. The radius is about 580 m and 870 m at 1 km and 2 km altitude, respectively. Thus, the retrieved bias can be ignored in this study."

4) Bottom of page 4. These jets are important as the shear associated with them is hypothesised to lead to the gravity waves. It is not clear what the cause of these jets is though. The big changes in wind direction are unusual. Can you offer any explanation? It is possible that the upper jet is some sort of nocturnal low level jet as it seems to strengthen during the night and weaken a little around dawn, but there is not much evidence of wind turning with time as one might expect to see if this was an inertial

oscillation. Have you tried looking at hodographs at a specific height from the lidar winds to see evidence of this? Are there other explanations, e.g. the height of the surrounding topography leading to channelling in the valley?

**Response**: Thanks. During the field experiments, it seems that most of these jets are nocturnal low level jet from Fig. A2-3. The changes in wind direction of ~90° are not unusual in Fig. A3. It is difficult to see if this was an inertial oscillation with a change in wind direction of ~90° for us. The surrounding topography, e.g., Dabie Mountain and Mountains around Huangshan, and weather system, e.g., Typhoon passed by, may have an effect on this change in wind direction. However, these question are beyond our scope of this study. If someone is interested in this subject, future studies of these jets and cooperation may be helpful after the peer-review completion.

5) Page 6, line 2. It seems likely that this aerosol concentration profile is associated with a stably stratified boundary layer, but I'm not sure that you can definitively say this from the lidar observations since you have no direct measure of temperature or stability.

**Response**: Thanks. We changed this description in the revised manuscript.

**Changes**: Page 6, line 2 to line 3. "Thus the ABL seems to be stably stratified as the CNR may represents the aerosol concentration in some cases."

6) Page 7, figure 5. It would be useful to include a plot of the Scorer parameter here in addition to N2 since it is hypothesised that wind shear is important.

**Response**: Thanks for this suggestion. In Fig. 10, we have plot the vertical profile of vertical wave number squared, which is also associated with wind shear as shown in Taylor-Goldstein equation (Eq. 4). The Scorer parameter $l$ is given by the following equation:

$$l^2 = \frac{N^2}{U^2} + \frac{\bar{u}_{zz}}{U} \qquad\qquad (R1)$$

where $U$ is the vertical profile of the barrier normal component of the horizontal wind, and other parameters are defined in the manuscript. The dominant first two terms on the right-hand side in Eq. 4 are similar to those two terms on the right-hand side in Eq. R1, except that intrinsic phase speed $c_i$ is replaced by horizontal wind speed $U$. The role of wind shear has been discussed in the Discussion with the Taylor-Goldstein equation.

7) Page 7, lines 17-18. I am not sure you can say anything about the presence of absence of waves above 1800-2000m. The lidar barely extends above 2000m between 0400-0900 and the CNR is quite low, with noisy w fields. I would reword this conclusion.

**Response**: Thanks. We delete this description in the revised manuscript.

8) Page 8, Figure 6. If you are plotting W1 and W2 in the text, then you need to at least briefly explain them here. The main text should be understandable without reading the appendix. The appendix is for additional details. At the moment the figure is meaningless without knowing what it means having only W1 or only W2. (Actually this is still not very clear, even after reading the appendix – see below).

**Response**: Thanks. We are regret for the missing description of W1 and W2 here in the

manuscript. W1 and W2 are the potential waves identified by LS spectral and Morlet wavelet spectral, respectively. Only W1 (W2) means the potential waves are only identified by LS (Morlet wavelet) spectral method, and not identified by Morlet wavelet (LS) spectral method, simultaneously.

During the revision, we reconsidered this part of GW identification. We present this part to demonstrate the relation between GW occurrences and background wind. However, on the one hand, this relation has been studied in several previous studies (Mahrt, 2014) and can be also analyzed in Sect. 4.2. On the other hand, the method if GW identification described in Appendix B is simple and rough, not serious as discussed in Appendix B. Therefore, we decided to delete these text and figure related to GW identification during the whole field campaign in the revised manuscript. The same to the Appendix B. This deletion will not affect the analysis and result of this work. At the same time, this deletion will also help us concentrate on the analysis of the unique long-lived ducted GW.

**Changes**: Appendix B is deleted. Sect. 3.2 is reorganized as follows:

"There are complex relationships between GWs and background wind conditions. Submeso wavelike motions, that any nonturbulent motions on horizontal scales smaller than 2 km and with periods of tens of minutes, are primarily generated in very weak winds in nocturnal boundary layer (Mahrt, 2014). Noted that the wind speed from 4 to 5 in September 2018 are weakest during the whole field experiment in Fig. A2. In order to understand the relationship between this ducted GW and background wind, a temporal spatial window of 1-hour length and 200-m height, and shifted in steps of 1 hour temporally and 100 m vertically is used. Mean horizontal wind speed and wind direction in each window during the whole field campaign in Anqing are easily obtained. The wind rose of the horizontal wind during the field experiment is shown in Fig. 6. It is apparent that northeasterly wind and southwesterly wind are prevailing around the station in the ABL during the whole field campaign. The infrequently observed ducted GWs in Fig. 2 occur accompanying infrequently westerly weak wind. It is interesting to note that the long-narrow plain area along Yangtze River around Anqing between Huangshan and Dabie Mountain is also along the direction of northeast-southwest as shown in Fig. 1a. The typical elevations of Huangshan and Dabie Mountain are about 1~2 km. Strong wind along northwest-southeast direction may be blocked in the ABL, thus leading to the weak wind along northwest-southeast direction after the wind flowing over Huangshan or Dabie Mountain and the prevailing wind along northeast-southwest direction. As GWs are favour to generate in weak wind conditions. we can imagine that Dabie Mountain and Huangshan may have an impact on GWs in Anqing. However, surrounding hills around the station as shown in Fig. 1b may also affect the generation and existence of GWs. The effect of surrounding hills will be studied by numerical simulations in next section."

9) Page 8, Figure 6. Also, are these plots for the whole field campaign or just the case study on 4th - 5th Sept. What height did you go up to? How was this chosen?

**Response**: These plots are for the whole field campaign in Anqing. The height range is from 100 m to 2000 m above ground. In the raw manuscript, a temporal spatial window

of 4-hour length and 200-m height, and shifted in steps of 1 hour temporally and 100 m vertically is employed. The window is utilized only when valid data covers greater than 80% of the window. In this condition, GWs can be roughly identified with the method described in Appendix B. Therefore, the height is decided by the carrier to noise ratio (CNR) of the raw data. As responded in major comment 8, Fig. 6 and related text are deleted to concentrate on the analysis of the unique long-lived ducted GW.

**Changes**: Page 8, line 8 to line 9. "Mean horizontal wind speed and wind direction in each window during the whole field campaign in Anqing are easily obtained."

10) Page 9, lines 15-16. This is a very bold claim! Both the terms "mesoscale models" and "CFD models" can cover a wide range of different things. Ultimately both are solving the same equations. Which is better will depend on the details of individual models and their numerics, and the setup of individual simulations (e.g. resolution, turbulence scheme etc.). CFD models traditionally do not include many atmospheric processes, although the distinction is increasingly blurred. All your advantages could equally apply to a high resolution atmospheric model. I would just remove this sentence altogether.

**Response**: We agree that both mesoscale models and CFD models are solving the same equations. The mesoscale numerical model considers the multi-physical process (e.g., wind, temperature, humidity, water vapor). However, mesoscale model seams too coarse for analyzing microscopic terrain. In contrast, the CFD model makes up for the inadequacies of mesoscale with high resolution. In addition, CFD mode has the ability to capture micro turbulence structures. Most mesoscale models have coarser net grids than CFD models employed in this work. We delete this sentence in the revised manuscript.

11) Page 9, section 4.1. So which CFD model are you actually using? Is this a commercial code? Include a reference to the actual model and its validation if at all possible.

**Response**: The Renormalization group k-epsilon (RNG k-ε) model is employed in this work, which is proposed by (Yakhot et al., 1992). A more comprehensive description of RNG k-ε theory and its application to turbulence can be found in (Orszag, 1993). The RNG k-ε models products reliable predictions for wind flows over hilly terrain (Kim et al., 2000), are more computationally efficient than the Large Eddy Simulation (LES) models. The RNG k-ε model is a quite mature model which is widely verified in simulation of wind flow over complex terrain in recent years (El Kasmi and Masson, 2010; Yan et al., 2015).

The RNG k-ε turbulence model based on OpenFOAM is used in this work. OpenFOAM is the leading free, open source software for computational fluid dynamics (CFD), owned by the OpenFOAM Foundation and distributed exclusively under the General Public Licence (GPL). The GPL gives users the freedom to modify and redistribute the software and a guarantee of continued free use, within the terms of the licence. Open-source CFD tool is a more powerful research tool in comparison to proprietary software because of its flexibility to incorporate new implementation of field calculation and

post-processing, attracts users from both industry and academia. The details of the RNG k-ε model and core codes can be found in the following website: https://www.openfoam.com/documentation/guides/latest/api/classFoam_1_1RASMod els_1_1RNGkEpsilon.html.

**Changes**: Page 9, line 14 to 19. "In this study, a two-equation RANS model based on renormalisation group (RNG) methods is used to simulate wind field. The RNG k-ε model was developed to renormalize the Navier-Stokes equations which are account for the effects of smaller scale motions (Yakhot et al., 1992). The RNG k-ε model is a quite mature model which is widely verified in simulation of wind flow over complex terrain in recent years (El Kasmi and Masson, 2010; Yan et al., 2015). The RNG k-ε turbulence model used in this work is based on OpenFOAM. OpenFOAM is the leading free, open source software for CFD simulations."

12) Page 10, lines 13-14. With the first cell height at 5m then you are not resolving the roughness sub-layer at all, and so you need to apply some sort of wall function / Monin-Obukov similarity function rather than just the no-slip boundary condition. Are you doing this? If so, what?

**Response**: For a fully rough surface the roughness length $z_0$ and the roughness height $k_s$ are related via $k_s = z_0 e^{KB}$, where K is the von Karman constant (K≈0.4) and B≈8.5 is the constant in the logarithmic velocity profile for rough surfaces (Franke et al., 2004). $z_0$ is set as 0.15 for the landforms in this paper. After grid independence verification, the first layer height is 5.0 m for considering a wall function, when the resolution of the terrain is 20 m. A good result was obtained with almost the same mesh set as in this study, indicating that the present mesh generation is reasonable (Ren et al., 2018).

A rough-wall function was adopted, of which the formula is as follows (Ren et al., 2018):

$$\frac{u}{u^*} = \frac{1}{K} In(\frac{Ez_c}{Ck_s}) \qquad (R2)$$

where E = 9.793 is the wall constant, C = 0.327 is a roughness constant, $z_c$ is the distance to the cell center of the first wall adjacent cell, u is the velocity in the cell center, $u^*$ is the friction velocity.

**Changes**: Page 10, line 5 to 9. "A rough-wall function is adopted, of which the formula is as follows (Ren et al., 2018):

$$\frac{u}{u^*} = \frac{1}{K} In(\frac{Ez_c}{Ck_s}) \qquad (4)$$

where E=9.793 is the wall constant, C = 0.327 is a roughness constant, K≈0.4 is the von Karman constant, $k_s$ is the roughness height, $z_c$ is the distance to the cell center of the first wall adjacent cell, $u$ is the velocity in the cell center, $u^*$ is the friction velocity."

13) Presumably there is a prognostic equation for potential temperature or similar, in order to include stability effects? No mention of this. What boundary conditions are used for this variable?

**Response**: The CFD cases conducted in this study are used to reveal the influence of topography and wind shear on the generation of the persistent GWs. The thermal field is assumed to be uniform on horizontal direction. Temperature profiles from radiosonde

on 5 September is applied in this model. For this ducted GWs, buoyant flows are developed with low velocity and small temperature variations. As a result, the Boussinesq model is used in study, which considers only the effect of buoyancy in gravity terms. The Boussinesq approximation can be used instead of a constant density. This model treats density as a constant value $\rho_{ref}$ in all solved equations, except for the gravity and buoyancy term. The density is approximated as:

$$\rho = \rho_{ref} - \rho_{ref}\beta(T - T_{ref}) \tag{R3}$$

where $\beta$ is the thermal expansivity, $T_{ref}$ is a reference temperature. the couple of thermal and velocity is realized by air density which the function with temperature.

**Changes**: Page 10, line 15 to 18. "The CFD cases conducted in this study are used to reveal the influence of topography and wind shear on the generation of the persistent GWs. The thermal field is assumed to be uniform in a horizontal plane. Temperature profile from radiosonde on 5 September is applied in this model. In this work, buoyant flows are developed with low velocity and small temperature variations. As a result, the Boussinesq model is used in this work, which considers only the effect of buoyancy in gravity terms. The Boussinesq approximation can be used instead of a constant density. This model treats density as a constant value $\rho_{ref}$ in all solved equations, except for the gravity and buoyancy term in the momentum equation. The density $\rho$ is approximated as:

$$\rho = \rho_{ref} - \rho_{ref}\beta(T - T_{ref}) \tag{5}$$

where $\beta$ is the thermal expansivity, $T_{ref}$ is a reference temperature."

14) Page 10, line 9. What do you mean by "symmetric condition" at the upper interface? Which variables does this apply to?

**Response**: The actual conditions of upper interface are difficult to obtain. The upper interface of the computational domain are external boundaries representing the far fields of flow. If a constant pressure is applied in these boundaries, this may alter the inlet wind profile in case the prescribed pressure is not matched with the boundary velocity (Luketa-Hanlin et al., 2007). Symmetry condition can be used at the top boundaries to reserve the wind profile and eliminate the effect of changing the inlet profiles. On this condition, zero Gradient is set for all vertical physical variables, and the vertical velocity is set as zero (Tran et al., 2019). In addition, symmetric condition facilitates computational convergence. Therefore, the upper interface is assumed as symmetric condition, which facilitates computational convergence.

**Changes**: Page 9, line 31 to page 10, line 1. "On this condition, zero gradient is set for all vertical physical variables, and the vertical velocity is set as zero."

15) Page 11, lines 17-18. I don't understand the comment that the flow solution is initiated as a steady state in all cases except case 7 and 8. How can you initiate at a steady state? Do you mean you initialise with the merged wind profile everywhere? In general this will not be an exact solution to the model equations, even in 1-d, so this isn't a steady state. Even if it was a solution in 1-d, the inclusion of topography means there will be variations across the domain which you cannot know without solving first.

I assume the additional complication for cases 7 and 8 is that the addition of a constant velocity everywhere would break the no-slip boundary condition. How do you deal with that? Is the velocity near the surface reduced to produce a consistent initialisation / inlet profile? If so, how?

**Response**: We do not mean initialise with the merged wind profile everywhere. The initial wind profile everywhere is set to be zero, except inlet. The steady state means the state when the turbulence developed fully, i.e., when the wind-inlet passed by the ABL and varies regularly above lidar station. The inlet profile is consistent. In case 7 and 8, the turbulences do not develop fully until the time of ~2 hour and ~1 hour, respectively. After that time in case 7 and 8, and after time of 0 in all other cases, the flow dynamics vary regularly as shown in Fig. 9 in raw manuscript.

**Changes**: Page 11, line 10 to line 14. "It should be noted that the time of 0 represents a steady state in all cases except case 7 and 8. Here, the steady state means the state when the turbulence developed fully, i.e., when the wind-inlet passed by the ABL and varies regularly above lidar station. In case 7 and case 8, the time of 0 is defined as when the simulations started running and the velocity-inlet flowed from the west boundary at the same time."

16) Page 11, table 2. You don't actually define u0. I assume this is the merged wind profile (lines 4-6), in which case define it there. It would be useful to have a plot of this model wind profile, and also the potential temperature profile.

**Response**: Yes, $u_0$ is the merged wind. This has been pointed in the revised manuscript.

**Changes**: Page 10, line 24 to line 25. "The 1-hour mean zonal wind under 2 km from lidar and zonal wind above 2 km from ERA5 reanalysis data at 00:00 in 5 September 2018 are merged as the sustained import wind profile $u_0$ in the west boundary of the computational domain."

17) The simulations presented are all 2-D, however the analysis elsewhere suggests that topography might be constraining the low level flow. Why choose to conduct 2-D simulations? What is the impact of this? This choice needs to be justified, and the limitations discussed.

**Response**: When simulating wind flow in complex mountain areas, more accurate results can be obtained by using 3D model with accurate boundary conditions. However, the actual wind field and terrain are complex. It is difficult to obtain accuracy boundary conditions for 3D model. In addition, 3D model consumes much more computing resources and time than 2D model.

As a simplification of the actual mountain model, the comparison between numerical simulation results and field experiments shows that the two-dimensional model can simulate the actual topographic flow well (Miller and Davenport, 1998; Toparlar et al., 2017; Walmsley et al., 1984). Some basic theories and empirical formulas of complex mountain wind field are built on the basis of two-dimensional model. Therefore, the two-dimensional terrain simulation of mountain wind field has a wide range of theoretical significance and practicability.

**Changes**: Page 16, line 11 to line 19. "More accurate results can be obtained by using

three-dimensional (3D) model with accurate boundary conditions when simulating wind flow in complex mountain areas. However, the actual wind field and terrain are complex. It is difficult to obtain accuracy boundary conditions for 3D model. In addition, 3D model consumes much more computing resources and time than 2D model. As a simplification of the actual mountain model, the comparison between numerical simulation results and field experiments shows that the two-dimensional model can simulate the actual topographic flow well (Miller and Davenport, 1998; Toparlar et al., 2017; Walmsley et al., 1984). Some basic theories and empirical formulas of complex mountain wind field are built on the basis of two-dimensional model. Therefore, the two-dimensional terrain simulation of mountain wind field has a wide range of theoretical significance and practicability. By using this simplified 2D model, the influence of terrain on GWs can be still analyzed."

18) Page 13, lines 15-16. If these waves were topographically generated, one would expect stationary waves rather than the propagating waves seen here. You have only shown time-height plots. If you look at vertical cross sections is there evidence of stationary topographically generated gravity waves at all?
**Response**: We have shown cross sections of vertical wind and zonal wind in the whole 2-D simulation domain in the video supplement (https://doi.org/10.5446/41847). In this movie, some quasi-stationary topographically generated GWs can be found around the east (right) hill.

19) While there are a relatively comprehensive set of simulations presented here, there is relatively little attempt to explain physically why the differences between the simulations occur. It might be fruitful to look at the Scorer parameter for different wind profiles. Does this explain the differences in wave trapping observed for example? One factor which is not investigated at all is the role of stability. Without stability there would be no waves at all, and the presence of these trapped horizontal wave modes must be at least partly due to the low level inversion. How would altering this affect the results?
**Response**: The temperature profiles are the same among the 16 cases. The stabilities will only vary with vertical profiles of horizontal wind in Scorer parameter as shown in Eq. R1. We agree that partly of the trapped horizontal wave modes due to the inversion layer, which has been discussed in Fig. 10 in raw manuscript. "The thermal duct is dominant under the temperature inversion with maxima buoyancy frequency squared for all propagation directions as shown in Fig. 5. The Doppler duct is dominant between ~0.5 and ~1.5 km altitude range due to the critical level induced by the low-level jet of wind maximum in a particular direction."

20) Page 13, line 22. It is rather unusual to see negative values of N at 2km. This implies an unstable atmosphere at this height. Do you have any idea what is driving this? I note that these negative values are fairly small, and only over a narrow height range (Figure 5b). It is interesting to note that the waves appear to reach well above the stable boundary layer in this case (perhaps 200-300m depth), even though there are several

near-neutral levels below 2km. Can you explain this? Plotting temperature profiles (rather than potential temperature) and the choice of scales for the N2 plot makes this difficult to judge though.

**Response**: The negative values of $N^2$ may be due to the rapid decrease of temperature with altitude as shown in Fig. 5a. This may be relation to the ABL top. The temperature, water vapor and aerosols are mixed better under this top due to stronger turbulence than above this top. It is a result of the combination of thermal (stable boundary layer) and Doppler ducts. The Doppler duct play a dominant role above the stable boundary layer as discussed in the Discussion. Potential temperature profiles are added in the revised Fig. 5.

21) Appendix B. From what is written I cannot see why you need to choose and compare two separate wave frequencies W1 and W2. From figure B1 I am guessing that W1 is the most significant peak with LS and W2 the most significant peak with the Morlet wavelet? The text does not explicitly say this. Similar, I am assuming from the figure that the wave is identified as the most significant peak, although this is not explicitly stated. Only when similar waves are identified by both methods is the case treat as a GW. Why do you use this criterion? This whole section could be better explained.

**Response**: Yes, the most significant wave is identified by compare W1 and W2. If a wave is significant in a temporal window, this wave can be identified by different method. Some turbulence or noise may be treated as waves with only one method. However, this method is simple and rough, not serious. As responded in major comment 8, we deleted this section and related part in the text in the revised manuscript in order to avoid confusing readers.

[Figure]

**Figure R1**. The same to Fig. B1 in the raw manuscript. But (d-f) are replaced by false detection.

22) Page 22, line 1. Do you have any evidence of false detection? Are any of the wave signals unrealistic? There doesn't seem much evidence to confirm or deny this at the moment so it is rather speculative.

**Response**: Yes, we have. A false detection is shown in Fig. R1d-f. Though with some false detection, we thought the statistical results are still useful to enrich our knowledge of such waves as discussed in second paragraph in Appendix B. As responded in major comment 8, we deleted this section and related part in the revised manuscript.

**References**

Birch et al (2013) Impact of soil moisture and convectively generated waves on the initiation of a West African mesoscale convective system. QJRMS 139 1712-1730.

Grubisic et al (2008) THE TERRAIN-INDUCED ROTOR EXPERIMENT: A Field Campaign Overview Including Observational Highlights. BAMS 89 1513–1534.

Lac, Lafore and Redelsperger (2002) Role of Gravity Waves in Triggering Deep Convection during TOGA COARE. JAS 59 1293–1316.

Lapworth and Osborne (2016) Evidence for gravity wave drag in the boundary layer of a numerical forecast model: A comparison with observations. QJRMS 142 3257–3264.

Marsham and Parker (2006) Secondary initiation of multiple bands of cumulonimbus over southern Britain. II: Dynamics of secondary initiation. QJRMS 132 1053-1072.

Tsiringakis, Steenveld, Holtsag (2017) Small-scale orographic gravity wave drag in stable boundary layers and its impact on synoptic systems and near-surface meteorology. QJRMS 143 1504–1516.

**Minor comments:**

Title. "Long-lived High Frequency Gravity Waves in the Atmospheric Boundary Layer" would be better English.
**Response**: Changed as suggested.

Page 4, line 4. "Wind field"
**Response**: Corrected.

Page 4, line 20. "4-dimensions"
**Response**: Corrected.

Page 6, Fig 3 caption. Should this be "Cone-of-Influence"?
**Response**: Corrected.

Page 7, line 15. Units should not be in italics.
**Response**: Corrected.

Page 9, line 2. The word "quadrature" is not really appropriate here. I would say "is perpendicular to the corresponding wind rose". This occurs at other places in the text too.
**Response**: Corrected.

Page 9, line 25. "simulate the wind field".
**Response**: Corrected.

Page 12, figure 9 caption. "wave motions are not only exist in the vertical wind". "no wave motions are generated". "cases 2-4"
**Response**: Corrected as suggested.

Page 15, line 18. ". . . perturbations of the GW were 90∘ out of phase with vertical perturbations..."
**Response**: Corrected as suggested.

**References**:

El Kasmi, A., and Masson, C.: Turbulence modeling of atmospheric boundary layer flow over complex terrain: a comparison of models at wind tunnel and full scale, Wind Energy, 13, 689-704, 10.1002/we.390, 2010.

Franke, J., Hirsch, C., Jensen, G., Krüs, H. W., Miles, S. D., Schatzmann, M., Westbury, P. S., Wisse, J. A., and Wright, N.: Recommendations on the use of CFD in wind engineering, Proceedings of the International Conference on Urban Wind Engineering and Building Aerodynamics, 2004.

Kim, H. G., Patel, V. C., and Lee, C. M.: Numerical simulation of wind flow over hilly terrain, Journal of Wind Engineering and Industrial Aerodynamics, 87, 45-60, 10.1016/s0167-6105(00)00014-3, 2000.

Luketa-Hanlin, A., Koopman, R. P., and Ermak, D. L.: On the application of computational fluid dynamics codes for liquefied natural gas dispersion, Journal of hazardous materials, 140, 504-517, 10.1016/j.jhazmat.2006.10.023, 2007.

Mahrt, L.: Stably Stratified Atmospheric Boundary Layers, Annual Review of Fluid Mechanics, 46, 23-45, 10.1146/annurev-fluid-010313-141354, 2014.

Miller, C. A., and Davenport, A. G.: Guidelines for the calculation of wind speed-ups in complex terrain, Journal of Wind Engineering and Industrial Aerodynamics, 74-76, 189-197, 10.1016/s0167-6105(98)00016-6, 1998.

Orszag, S. A.: Renormalisation group modelling and turbulence simulations, Near-wall turbulent flows, Tempe, Arizona, 1993.

Ren, H., Laima, S., Chen, W.-L., Zhang, B., Guo, A., and Li, H.: Numerical simulation and prediction of spatial wind field under complex terrain, Journal of Wind Engineering and Industrial Aerodynamics, 180, 49-65, 2018.

Toparlar, Y., Blocken, B., Maiheu, B., and van Heijst, G. J. F.: A review on the CFD analysis of urban microclimate, Renewable and Sustainable Energy Reviews, 80, 1613-1640, 10.1016/j.rser.2017.05.248, 2017.

Tran, V., Ng, E. Y. K., and Skote, M.: CFD simulation of dense gas dispersion in neutral atmospheric boundary layer with OpenFOAM, Meteorology and Atmospheric Physics, 10.1007/s00703-019-00689-2, 2019.

Walmsley, J. L., Taylor, P. A., and Salmon, J. R.: Simple guidelines for estimating windspeed variations due to small-scale topographic features–an update, Climatological bulletin, 23, 3-14, 1984.

Yakhot, V., Orszag, S., Thangam, S., Gatski, T., and Speziale, C.: Development of turbulence models for shear flows by a double expansion technique, Physics of Fluids A: Fluid Dynamics, 4, 1510-

1520, 1992.

Yan, B. W., Li, Q. S., He, Y. C., and Chan, P. W.: RANS simulation of neutral atmospheric boundary layer flows over complex terrain by proper imposition of boundary conditions and modification on the k-ε model, Environmental Fluid Mechanics, 16, 1-23, 10.1007/s10652-015-9408-1, 2015.

---

## Author Comment (AC2) · 30 Sep 2019

We appreciate all the great efforts and constructive comments from the reviewers. We have revised the manuscript carefully according to the reviewers' comments and suggestions. Our point-by-point responses and changes are appended below in blue fonts.

**Anonymous Referee #2**

The manuscript presents the results of lidar observations of wave like variations of wind velocity vector vertical and horizontal components in the boundary layer of atmosphere obtained during the field experiment in August-September 2018 in the location of Anqing, China. The experimental results are accompanied by the results of model numerical simulation of the wind field disturbed by the topographical objects. Based on the obtained experimental and computational results the conclusions about mechanism of generation of wave variations of wind velocity are formulated in the manuscript.

**Major comments:**

1) The general goal of the study is not clear from the manuscript. It may be proposed that this goal is study of atmospheric waves arising under conditions of stable thermal stratification in the boundary layer of atmosphere. But only one event of atmospheric wave on 4-5 September is analyzed in the manuscript with the use of information about the profile of temperature in height. Moreover, even for that event there is no data on temperature profile measured with the radiozonde at 19:15 on 4th September in the manuscript. To improve understanding of this issue, it may be useful to present the temperature profiles in height during all the period of field experiment and carry out the analysis of frequency of wave events not only with taking into account the magnitudes of wind velocity and wind shear, as it is presented in Fig. 6, but also with consideration of the temperature stratification.

[Figure]

**Figure R1.** The histograms of GWs occurrence with squares of buoyancy frequency (upper panel). GWs occurrence rate versus buoyancy frequency (bottom panel).

**Response**: We revised the manuscript to make the goal of this study more clearly. The

temperature profile measured with the radiosonde at 19:15 on 4th September in the revised Fig. 5. We had considered the temperature stratification $N^2$ before submit the manuscript. The result shows that there is not any significant relation between the frequency of wave events and temperature stratification as shown in Fig. R2.

However, the method of GW identification is simple and rough, not serious. As replied in major comment 8 to **Anonymous Referee #1**, we deleted the related text of GW identification and frequency of wave events in the revised manuscript due to following reason:

"During the revision, we reconsidered this part of GW identification. We present this part to demonstrate the relation between GW occurrences and background wind. However, on the one hand, this relation has been studied in several previous studies (Mahrt, 2014) and can be also analyzed in Sect. 4.2. On the other hand, the method if GW identification described in Appendix B is simple and rough, not serious as discussed in Appendix B. Therefore, we decided to delete these text and figure related to GW identification during the whole field campaign in the revised manuscript. The same to the Appendix B. This deletion will not affect the analysis and result of this work. At the same time, this deletion will also help us concentrate on the analysis of the unique long-lived ducted GW."

2) The representativeness of the estimates of the mean wind velocity. As mentioned in line 6 on p.4, measurement duration of radial velocity in one direction is 10 s during this experiment. For used in the experiment scanning geometry such duration of measurements is insufficient in order to obtain statistically justified estimates of the mean wind velocity components. Actually, it is well known that integral spatial scale of wind turbulence is proportional to the height under ground in the lower atmosphere and can reach a few hundreds of meters at the heights 600-2000 m. To obtain statistically justified estimate of the mean velocity, the velocity fluctuations caused by the turbulent inhomogeneities of all the scales up to hundreds of meters must be averaged. Even for observed in the experiment maximal velocity 10 m/s in order to average velocity fluctuations caused by the turbulent inhomogeneities of velocity field of such spatial scales it requires few hundreds seconds, at least.

**Response**: We totally agree that "To obtain statistically justified estimate of the mean velocity, the velocity fluctuations caused by the turbulent inhomogeneities of all the scales up to hundreds of meters must be averaged." We have checked the raw radial wind in both north beam and west beam. The ducted wave motions are still significant. In addition, turbulence activity is relatively weak in the nocturnal residual layer. As shown in Fig. 2, wave motions are significant without average, though smoothed perturbation will be better in Fig. 7f. Hence, the turbulence inhomogeneities and average will not affect the result of this work. Though the amplitude of zonal/meridional wave motions may be affected due to the turbulence and wave motions.

3) What is the reason of variations of wave period in Figs 3, 4? Model calculations in Fig. 8 do not reproduce wave period variations. It may be useful to compare the experimental and calculation results in more detail by combining the experimental and

calculated data in one plot. It is difficult to compare and understand the results in Figs. 8b, 8c.

**Response**: The variation of the wave period may be caused by turbulence and the change of background winds. The reason why model calculations do not reproduce wave period variation may be due to stable boundary condition in the model, while lidar observed wind may be affected by many reasons, such as terrain, weather system and so on. Thanks for the suggestion of the comparation. We think that a time-height cross section of vertical wind perturbation would be better with more information. Therefore, we added a panel of comparison of vertical wind perturbation at 1.0 altitude from both observation and simulation results in revised Fig. 8.

**Changes**: Fig. 8.

4) The code used for numerical modeling must be described in more detail. As it can be proposed, some version of the program CFD Fluent was used in the modeling. Accordingly to Eqs. (2), (3), it is required to set a lot of input turbulent parameters in order to perform the modeling using that code. None of these input parameters is determined experimentally. At least, there is no information about that in the manuscript. If so, there is no any base for quantitative comparison of the experimental and computational results and conclusions about the mechanism of wave generation.

**Response**: The description of numerical simulations is reorganized with more details in Sect. 4.1. The input turbulent parameters recommended by OpenFOAM are applied (https://www.openfoam.com/documentation/guides/latest/api/classFoam_1_1RASMo dels_1_1RNGkEpsilon.htm). The default model coefficients of RNG k-ε are: $G_{1\epsilon} = 1.42; G_{2\epsilon} = 1.42; G_{3\epsilon} = -0.33; \alpha_k = 1; \alpha_\varepsilon = 1.22$.

**Changes**: Page 9, line 26 to page 10, line 18. "In this paper, the input turbulent parameters recommended by OpenFOAM are applied. The default model coefficients of RNG k-ε are: $G_{1\epsilon} = 1.42; G_{2\epsilon} = 1.42; G_{3\epsilon} = -0.33; \alpha_k = 1; \alpha_\varepsilon = 1.22$.

To simplify the numerical simulation processes, a two-dimensional (2D) rectangle computational domain is applied in this study. with 70 km in horizontal and 5 km in vertical from sea level. The upper interface extended to 5 km is set as symmetric condition to prevent the influence of upper interface on the region concerned that below 2 km. On this condition, zero gradient is set for all vertical physical variables, and the vertical velocity is set as zero. The vertical height of the first layer of grid cells is 5 m. The spatial resolution is approximately 20 m in both horizontal and vertical. The total number of computational grid cells is 875,000. The velocity-inlet is westerly and constant in the west boundary of the computational domain. The easterly interface is set as pressure-outlet boundary to improve reversed flow. The topography is set as no-slip wall condition. A rough-wall function is adopted, of which the formula is as follows (Ren et al., 2018):

$$\frac{u}{u^*} = \frac{1}{K} In(\frac{Ez_c}{Ck_s}) \tag{4}$$

where $E$=9.793 is the wall constant, $C = 0.327$ is a roughness constant, $K≈0.4$ is the von Karman constant, $k_s$ is the roughness height, $z_c$ is the distance to the cell center of the first wall adjacent cell, u is the velocity in the cell center, $u^*$ is the friction velocity.

The simulation is run with a time step of 0.5 s.
The CFD cases conducted in this study are used to reveal the influence of topography and wind shear on the generation of the persistent GWs. The thermal field is assumed to be uniform in a horizontal plane. Temperature profile from radiosonde on 5 September is applied in this model. In this work, buoyant flows are developed with low velocity and small temperature variations. As a result, the Boussinesq model is used in this work, which considers only the effect of buoyancy in gravity terms. The Boussinesq approximation can be used instead of a constant density. This model treats density as a constant value $\rho_{ref}$ in all solved equations, except for the gravity and buoyancy term in the momentum equation. The density $\rho$ is approximated as:

$$\rho = \rho_{ref} - \rho_{ref}\beta\left(T - T_{ref}\right) \tag{5}$$

where $\beta$ is the thermal expansivity, $T_{ref}$ is a reference temperature."

**Minor comments:**
1) Temperature profile curves in Fig.5 should be identified.
**Response**: The temperature profile from radiosonde can be easily identified. We know that a shift of each temperature profile from ERA5 would be better. Nevertheless, the vertical structure of the temperature profiles can be still identified without a shift. An inversion layer is obvious under 0.5 km altitude.

2) Parameter N in line 14, p.7 should be expressed by formula.
**Response**: Added.

3) Resolution of wind and temperature experimental data in height should be indicated.
**Response**: Added.

4) Magnitudes of ăˇDOA, and ăˇDOB in Fig.8 and Table 2 must be indicated.
**Response**: It may be assumed that ăˇDOA (ăˇDOB) is $h_A$ ($h_B$).
**Changes**: Page 10, line 23. "The maximum elevation of A and B are approximately 250 m and 600 m, respectively."

**References**:
Mahrt, L.: Stably Stratified Atmospheric Boundary Layers, Annual Review of Fluid Mechanics, 46, 23-45, 10.1146/annurev-fluid-010313-141354, 2014.
Ren, H., Laima, S., Chen, W.-L., Zhang, B., Guo, A., and Li, H.: Numerical simulation and prediction of spatial wind field under complex terrain, Journal of Wind Engineering and Industrial Aerodynamics, 180, 49-65, 2018.

---

## Referee Report (RR1)

**Review of revised version of acp-2019-256. "Long-live High Frequency Gravity Waves in Atmospheric Boundary Layer: Observations and Simulations" by Jia et al.**

Summary: This revised version of the manuscript is a definite improvement and has addressed the majority of the reviewers' comments from last time. I think removing the wavelet analysis is probably wise and leads to a more focussed paper. I have a couple of minor comments still remaining.

Minor comments:

1) The written English throughout would benefit from some thorough proof reading by a native English speaker. Although I think the meaning is generally clear, I had to re-read a number of sentences to make sure I had the meaning right since they were oddly phrased. I haven't commented on these in detail since this is an editorial matter.

2) Reviewer 1, major comment 13. The amended text here is definitely clearer, however I am still not completely sure what is done. For instance, the comment that the thermal field is assumed to be uniform in a horizontal plane cannot be true. I think you mean that the reference temperature field is uniform in a horizontal plane. It is also not entirely clear whether $T_{ref}$ and $\rho_{ref}$ are constant, or functions just of height? As it stands the formulation does not seem entirely consistent. Normally in the Boussinesq approximation it is assumed that is $\rho_{ref}$ constant and that fluctuations in $\rho$ are neglected except in the buoyancy term. $T_{ref}$ (or more usually potential temperature $\theta_{ref}$ need not be constant, but is a function of height, with the buoyancy term written in terms of $T-T_{ref}$ or $\theta-\theta_{ref}$. If variations of $\rho_{ref}$ with height are included then this is typically the anelastic approximation.

3) Reviewer 1, major comment 13. The response does not discuss boundary conditions for T at all. Is it a fixed surface temperature or are there any imposed fluxes?

4) Reviewer 1, major comment 15. Maybe this is a misunderstanding due to language, but I don't see how you can have a steady state solution at t = 0 if you are initialising the model with zero velocity everywhere. There must be some spin up time for the mean flow and turbulence surely?

5) Reviewer 1, major comment 17. I don't think this is a very satisfactory response. Sure, 2-d simulations can be useful as an idealisation to study processes. Whether they are a good model for the real world depends a lot on the particular topography and how two-dimensional it is. As such, the references may be misleading since they are for different locations. There are plenty of idealised 3-d studies which show differences from idealised 2-d studies. My point is that if you are suggesting elsewhere that topography is constraining the flow in this case (a 3-d effect), then you should at least discuss the possible impact of only modelling 2-d flow for this site. How 3-d is it? Why did you choose the transect you did? How representative is this?

6) Fig 5, caption " between 22:00"

---

## Editor Decision (ED1)

p.17 l.11

The initial thermal field is assumed to be horizontally uniform, using the vertical  temperature profile from the radiosonde on 5 September. . The topography is set at  fixed temperature , since the heat flux  at the ground  was unavailable. In this work, buoyant flows are developed with low velocity and small temperature variations in each layer. The Boussinesq approximation is applied for each thin layer.  which treats density $\rho_{re}(z)$ as a constant value at altitude z in all solved equations, except for the gravity and buoyancy term in the momentum equation. The fluctuation of (z) is caused by temperature $T(z)$, neglecting the influence of pressure. The density (z) is approximated as:

$\rho(z)=\rho_{ref}(z)-\rho_{ref}(z)\beta(T(z)-T_{ref}(z))$ (5)

where $\beta$ is the thermal expansivity, and $T_{ref}(z)$ is the reference temperature at altitude z. The Boussinesq approximation is similar  to the anelastic approximation in this form. The main difference between the Boussinesq  and anelastic approximations is that the anelastic approximation considers the influence of both pressure and temperature in fluctuations of $\rho$. Considering computational convenience and convergence, the Boussinesq approximation is adopted in this work.

p. 18 l.9

Based on this result, wind profiles $u_z$ with different wind shear, and topography with different height of hills A and B are employed in the CFD numerical simulations. A detailed list of boundary conditions is presented in Table 2. The  simulated  zonal winds and vertical winds above the lidar for all cases are shown in Fig. 8. It should be noted that a time of 0 represents a  steady state in all cases except cases 7 and 8, not the real time after the simulations started running. Here,  we mean by a steady state that which pertains  when the  turbulence is fully developed. In cases 7 and  8, a time of 0 is defined as that when the simulations started running and the  input velocity flowed from the western boundary at the same time.

p.23 l.10

 A 2D model is used in this simulation rather than, a 3D model which in principle can more accurately simulate the atmospheric flow , Considering the direction of the low-level jet and the maximum background wind flow , this zonal transect in this case is appropriate  for a 2D model: The influence of terrain on atmospheric flow is mainly in this direction. However, the 2D model cannot simulate the information in another dimension, e.g., lateral flow around the hillside and the blocking effect of the low terrain on both sides, leading to  errors compared with a 3D model. Nevertheless,  comparison between  numerical simulation results and field experiments show that  two-dimensional model can simulate the actual topographic flow well in some cases (Miller and Davenport, 1998; Toparlar et al., 2017; Walmsley et al., 1984). Furthermore, some basic theories and empirical formulas of complex mountain wind field are built on the basis of a two-dimensional model. In addition, a 2D model consumes much less computing resources and time than a 3D model. Therefore, the two-dimensional terrain simulation of the mountain wind field has a wide range of theoretical significance and  applicability.  By using this simplified 2D model, the influence of terrain on GWs can be  analyzed.

---

## Author Response (AR2)

**Summary:**

This revised version of the manuscript is a definite improvement and has addressed the majority of the reviewers' comments from last time. I think removing the wavelet analysis is probably wise and leads to a more focused paper. I have a couple of minor comments still remaining.

5 **Response**: Thanks very much for your constructive comments and suggestions. Our point-by-point responses and changes are appended below in blue.

**Minor comments:**

10 1) The written English throughout would benefit from some thorough proof reading by a native English speaker. Although I think the meaning is generally clear, I had to re-read a number of sentences to make sure I had the meaning right since they were oddly phrased. I haven't commented on these in detail since this is an editorial matter.

**Response**: We tried our best to improve the English, including professional help from language editing

15 agency.

2) Reviewer 1, major comment 13. The amended text here is definitely clearer, however I am still not completely sure what is done. For instance, the comment that the thermal field is assumed to be uniform in a horizontal plane cannot be true. I think you mean that the reference temperature field is uniform in

20 a horizontal plane. It is also not entirely clear whether $T_{ref}$ and $\rho_{ref}$ are constant, or functions just of height? As it stands the formulation does not seem entirely consistent. Normally in the Boussinesq approximation it is assumed that is $\rho_{ref}$ constant and that fluctuations in $\rho$ are neglected except in the buoyancy term. $T_{ref}$ (or more usually potential temperature $\theta_{ref}$ need not be constant, but is a function of height, with the buoyancy term written in terms of $T-T_{ref}$ or $\theta-\theta_{ref}$. If variations of $\rho_{ref}$ with height are

25 included then this is typically the anelastic approximation.

**Response**: Thanks for this comment. The initial thermal field is assumed to be uniform in horizontal. The buoyant flows are developed with low velocity and small temperature variations in horizontal. In order to simplify the model and make it easier to converge, Boussinesq approximation is applied for

thin layers at different heights. In Boussinesq approximation, the fluctuation of ρ is mainly caused by temperature, where the influence of pressure is neglected. This approximation leads to an error in the order of 1% if the temperature differences are below 15 K for air (Ferziger and Perić, 2002). The vertical computational height is up to 5 km. The density variation with height cannot be ignored. Both the reference density and the reference temperature vary as a function of height in layers. The vertical profiles of the reference temperature and the reference density are retrieved from the radiosonde on 5 September. The corresponding vertical profile of potential temperature has been shown in Fig. 7b. Boussinesq approximation in this form is similar with anelastic approximation. The main difference between Boussinesq approximation and anelastic approximation is that anelastic approximation considers the influence of both pressure and temperature in fluctuations of ρ. It should be noted that anelastic approximation is more closely to the real state of the atmosphere. Considering computational convenience and convergence, we prefer to adopt the Boussinesq approximation in each layer. The relevant descriptions have been clarified, including the formulation and boundary conditions of temperature.

**Changes**: Page 11, line 8 to line 15. "The initial thermal field is assumed to be uniform in horizontal. The temperature profile from the radiosonde on 5 September is applied as inlet and outlet boundary in this model. The topography is set as fixed temperature wall, since heat flux on the ground is unavailable. In this work, buoyant flows are developed with low velocity and small temperature variations in each layer. Boussinesq approximation is applied for each thin layer. Boussinesq approximation treats density $\rho_{ref}(z)$ as a constant value at altitude z in all solved equations, except for the gravity and buoyancy term in the momentum equation. The fluctuation of $\rho(z)$ is caused by temperature $T(z)$, neglecting the influence of pressure. The density $\rho(z)$ is approximated as:

$$\rho(z) = \rho_{ref}(z) - \rho_{ref}(z)\beta\big(T(z) - T_{ref}(z)\big) \tag{5}$$

where $\beta$ is the thermal expansivity and $T_{ref}(z)$ is the reference temperature at altitude z. Anelastic approximation is similar with Boussinesq approximation in this form. The main difference between Boussinesq approximation and anelastic approximation is that anelastic approximation considers the influence of both pressure and temperature in fluctuations of $\rho$. Considering computational convenience and convergence, Boussinesq approximation is adopted in this work."

3) Reviewer 1, major comment 13. The response does not discuss boundary conditions for T at all. Is it a fixed surface temperature or are there any imposed fluxes?

**Response**: Thanks for this comment. As responded in comment 2, the inlet and outlet boundary is set as fixed temperature using the temperature profile measured by the radiosonde on 5 September. The topography is set as fixed temperature wall, since heat flux on the ground is unavailable.

**Changes**: Page 11, line 8 to line 10. "The temperature profile from the radiosonde on 5 September is applied as inlet and outlet boundary in this model. The topography is set as fixed temperature wall, since heat flux on the ground is unavailable."

4) Reviewer 1, major comment 15. Maybe this is a misunderstanding due to language, but I don't see how you can have a steady state solution at t = 0 if you are initialising the model with zero velocity everywhere. There must be some spin up time for the mean flow and turbulence surely?

**Response**: Yes, there is some spin up time for the turbulence and mean flow. For cases 7 and 8, the inlet wind under the height of 1 km are ~5 and ~10 m/s, respectively. The spin up time are approximately $\frac{38\,km}{5\,m/s} \times \frac{1\,m/s}{3.6\,km/h} \approx 2\,h$ and $\frac{38\,km}{10\,m/s} \times \frac{1\,m/s}{3.6\,km/h} \approx 1\,h$, respectively. There are significant changes of horizontal wind and vertical wind around time of 2 h for case 7 and 1 h for case 8 as shown in Fig. 8. After this spin up time, the atmospheric flow varies stably with mean low, wave motions and turbulence due to the fully developed turbulent activities. We defined this state as a steady state in the previous version of response. A description of stable state may be more appropriate here instead of steady state. The time of 0 is defined when such a stable state appears after the spin up time in all other cases, not the real time after the simulations started running. Thus we can focus on the potential wave motions with the same time range after the spin up time.

**Changes**: Page 12, line 3 to line 7. "It should be noted that the time of 0 represents a stable state in all cases except cases 7 and 8, not the real time after the simulations started running. Here, the stable state indicates the state when the atmospheric flow varies stably with men flow, wave motions and turbulence due to the fully developed turbulent activities, i.e., when the wind-inlet passed through the ABL and varies stably above the lidar station."

5) Reviewer 1, major comment 17. I don't think this is a very satisfactory response. Sure, 2-d simulations can be useful as an idealisation to study processes. Whether they are a good model for the real world depends a lot on the particular topography and how two-dimensional it is. As such, the references may be misleading since they are for different locations. There are plenty of idealized 3-d studies which show differences from idealised 2-d studies. My point is that if you are suggesting elsewhere that topography is constraining the flow in this case (a 3-d effect), then you should at least discuss the possible impact of only modelling 2-d flow for this site. How 3-d is it? Why did you choose the transect you did? How representative is this?

**Response**: Thanks for your comments. Comparing with 2D model, 3D model can simulate the atmospheric flow with complete information. The simulation results will be more accurate with reasonable boundary conditions in 3D model. In this study, the low-level jet and the maximum background wind are mainly in the zonal direction. The influence of terrain on atmospheric flow is mainly in this direction. This has been mentioned in Sect. 4.2. "The horizontal location of the domain is roughly along the zonal white dash dotted line in Fig. 1b. This is because the low-level jet and the background wind are mainly in the zonal (east-west) direction." Thus we choose this representative transect. However, the 2D model cannot simulate the information in meridional dimension, e.g., lateral flow around the hillside and the blocking effect of the low terrain on both sides, leading to additional errors compared with 3D model. Nevertheless, the additional errors are acceptable for studying GWs in the dominant direction.

**Changes**:

Page 11, line 23. "The influence of terrain on atmospheric flow is mainly in this direction."

Page 16, line 10 to line 23. "Comparing with 2D model, 3D model can simulate the atmospheric flow with complete information. The simulation results will be more accurate with reasonable boundary conditions in 3D model. Considering the direction of the low-level jet and the maximum background wind, this zonal transect is appropriate in this 2D model. The influence of terrain on atmospheric flow is mainly in this direction. However, the 2D model cannot simulate the information in another dimension, e.g., lateral flow around the hillside and the blocking effect of the low terrain on both sides, leading to

additional errors compared with 3D model. Nevertheless, as a simplification of the actual mountain model, the comparison between the numerical simulation results and field experiments shows that the two-dimensional model can simulate the actual topographic flow well in some cases (Miller and Davenport, 1998; Toparlar et al., 2017; Walmsley et al., 1984). Some basic theories and empirical formulas of complex mountain wind field are built on the basis of a two-dimensional model. In addition, 2D model consumes much less computing resources and time than 3D models. Therefore, the two-dimensional terrain simulation of the mountain wind field has a wide range of theoretical significance and practicability. The additional errors with only 2D model are acceptable for studying GWs."

6) Fig 5, caption "between 22:00".

**Response**: Corrected.

Ferziger, J. H., and Perić, M.: Basic Concepts of Fluid Flow, in: Computational Methods for Fluid Dynamics, Springer, Berlin, Heidelberg, 1-20, 2002.

[revised manuscript text omitted]